Citation: *Molecular Systems Biology* 9:660
www.molecularsystemsbiology.com

# Indirect and suboptimal control of gene expression is widespread in bacteria

Morgan N Price[1,*], Adam M Deutschbauer[1], Jeffrey M Skerker[2,3], Kelly M Wetmore[1,3], Troy Ruths[1], Jordan S Mar[2,3], Jennifer V Kuehl[1], Wenjun Shao[4] and Adam P Arkin[1,2,3,*]

[1] Physical Biosciences Division, Lawrence Berkeley National Lab, Berkeley, CA, USA, [2] Department of Bioengineering, University of California, Berkeley, CA, USA, [3] Energy Biosciences Institute, University of California, Berkeley, CA, USA and [4] Department of Molecular and Cell Biology, University of California, Berkeley, CA, USA
* Corresponding authors. MN Price or AP Arkin, Physical Biosciences Division, Lawrence Berkeley National Lab, 1 Cyclotron Road, Mailstop 955-512L, Berkeley, CA 94720, USA. Tel.: +1 510 643 3722; Fax: +1 510 486 6219; E-mail: morgannprice@yahoo.com or aparkin@lbl.gov

Gene regulation in bacteria is usually described as an adaptive response to an environmental change so that genes are expressed when they are required. We instead propose that most genes are under indirect control: their expression responds to signal(s) that are not directly related to the genes' function. Indirect control should perform poorly in artificial conditions, and we show that gene regulation is often maladaptive in the laboratory. In *Shewanella oneidensis* MR-1, 24% of genes are detrimental to fitness in some conditions, and detrimental genes tend to be highly expressed instead of being repressed when not needed. In diverse bacteria, there is little correlation between when genes are important for optimal growth or fitness and when those genes are upregulated. Two common types of indirect control are constitutive expression and regulation by growth rate; these occur for genes with diverse functions and often seem to be suboptimal. Because genes that have closely related functions can have dissimilar expression patterns, regulation may be suboptimal in the wild as well as in the laboratory.
*Molecular Systems Biology* 9:660; published online 16 April 2013; doi:10.1038/msb.2013.16
*Subject Categories:* chromatin & transcription; microbiology & pathogens
*Keywords:* bacterial evolution; gene regulation; optimal regulation

## Introduction

In bacteria, gene regulation is traditionally thought of as an adaptive or homeostatic mechanism that allows the cell to respond to changing metabolic conditions or to environmental stresses (e.g., Wall *et al*, 2004; Seshasayee *et al*, 2009). The underlying rationale is that proteins 'should' be made only when needed so as to conserve cellular resources or because the protein's activity is detrimental in other conditions. The classic example is the induction in *Escherichia coli* of the *lac* operon in response to lactose: the *lac* operon is required for growth on lactose, and the *lac* operon is very weakly expressed in the absence of lactose. If the *lac* operon is artificially induced in the absence of lactose by adding a non-metabolizable analog of lactose to the medium, then the expression of the *lac* operon reduces the growth rate. This reduction in the growth rate reflects the cost of producing useless proteins instead of useful ones (Stoebel *et al*, 2008) and also the detrimental activity of the LacY permease in some conditions (Eames and Kortemme, 2012). The relative reduction in *E. coli*'s growth rate due to producing useless proteins seems to vary across growth conditions, but under low-cost conditions, the cost is approximately the fraction of total protein that is useless (Shachrai *et al*, 2010).

Although many specific examples of gene regulation appear to be adaptive under laboratory conditions, it is not clear whether the regulation of the majority of genes is adaptive. Genome-wide studies in both bacteria and yeast have found little correlation between changes in expression and the importance of genes for fitness (Birrell *et al*, 2002; Giaever *et al*, 2002; Smith *et al*, 2006; Deutschbauer *et al*, 2011). In other words, most genes are not downregulated when they are not needed for growth, and conversely, most genes that are upregulated do not seem to be important for fitness. This is surprising because under a cost-benefit model of optimal expression (Dekel and Alon, 2005), the optimal expression level of a gene will be much lower if there is little or no benefit (or fitness advantage) than if there is a large benefit. Thus, there is a puzzle as to why adaptive regulation does not seem to be more widespread in bacteria.

There have been several proposals for why genes might be expressed when they are not needed for fitness or why they might not be induced when they are needed. More precisely, these theories try to explain why bacteria with apparently non-adaptive regulation have not been outcompeted by other bacteria with more optimal regulation. First, some genes might be expressed in 'standby mode' because they will help the bacterium survive if conditions change (Fischer and Sauer, 2005). Standby control can be thought of as a way to reduce the delay inherent in adaptive control. If a gene is under adaptive control and is not expressed at all when it is not needed, then after conditions change and it becomes needed, there is a delay

until enough of the protein is produced to adapt to this new condition. During this delay, the cell might stop growing or might even die. Thus, uncertainty about the near future implies some possibility of a benefit from expressing a gene that is not currently needed. If there is a significant chance of obtaining a benefit in the future, then the average future benefit will exceed the (certain) cost of expressing unneeded protein, so the optimal expression level will be above zero even though the gene currently confers no benefit. Conversely, if the gene is currently needed but conditions might change in the near future, this reduces the expected benefit of high expression, and hence reduces the optimal expression level. In other words, optimal standby control should dampen the dynamic range of expression without changing the pattern. (For a detailed example, see Supplementary Figure 1). Thus, optimal standby control cannot explain why there is so little correlation between relative expression (i.e., when genes are upregulated) and mutant fitness (i.e., when they are needed for optimal growth).

A second and related theory is that proteins that are only needed in small amounts might be expressed constitutively because the cost of adaptive control, such as the cost of making transcription factors, might exceed the benefit of making less of the protein when it is not needed (Wessely *et al*, 2011). The cost of regulation seems small—e.g., the LacI repressor is present at only 20–50 copies per cell (Milo *et al*, 2010)—so this theory should only apply to weakly expressed genes that have a low cost of unnecessary expression.

A third theory related to changing conditions is that microorganisms might use one environmental signal to 'anticipate' another (Tagkopoulos *et al*, 2008; Mitchell *et al*, 2009). Here, the change in environment is (somewhat) predictable, rather than being entirely random. For example, for a gut bacterium like *E. coli*, a rise in temperature might indicate that it has been ingested and will soon reach an anaerobic environment (Tagkopoulos *et al*, 2008), so genes for anaerobic respiration might be induced even though they are not immediately useful. It is not clear whether anticipatory control of expression is widespread in bacteria.

Fourth, horizontally transferred genes, which are common in bacteria, might lack regulation because of insufficient time to evolve appropriate regulation in their current host (Lercher and Pal, 2008). However, only the most recently transferred genes seem to lack regulation (Lercher and Pal, 2008). Also, regulation can evolve quickly (Stone and Wray, 2001; Berg *et al*, 2004), regulation can be conserved across transfer events (Price *et al*, 2008), and many horizontally transferred genes are under complex control by multiple transcription factors (Price *et al*, 2008). Thus, we doubt that horizontal gene transfer could explain why there is little correlation between relative expression (i.e., regulation) and mutant fitness genome-wide.

Fifth, the regulation of some genes might be suboptimal or maladaptive because the expression patterns of those genes are not under strong selection. More precisely, if altered regulation improves relative growth by less than $1/N_e$ per generation, where $N_e$ is the effective size of the bacterial population and the effect on growth is averaged across natural environments, then this altered regulation is unlikely to take over the population. Selectively neutral evolution could also

account for some of the complexity of gene regulation (Lynch, 2007). However, both regulatory sites (McCue *et al*, 2002; Rajewsky *et al*, 2002;) and the coexpression of genes (Price *et al*, 2007) are usually conserved between closely related bacteria, which implies that the regulation of most genes is under some selection. Furthermore, in *E. coli*, over half of all genes are present at above 0.1 mRNA per cell in a single condition, which corresponds to 30–60 proteins per cell (Lu *et al*, 2007) or over 1 in 100 000 of all protein molecules in the cell (Milo *et al*, 2010). Because the fitness cost of unnecessary expression of a gene is probably at least as great as its proportion of total protein, this implies that the fitness cost of unnecessary expression of the typical gene is at least $10^{-5}$ per generation. This is about the same as the estimated fitness cost of mutations in codon usage that are under selection (Bulmer, 1991). Thus, unnecessary expression of the typical protein should be under selection.

Finally, we propose that non-adaptive regulation is widespread in bacteria, at least in laboratory settings, because of two major factors. First, bacterial genomes encode far more operons than regulators. In the typical bacterium, only 4.2% of proteins are predicted to be transcription factors (Charoensawan *et al*, 2010). With so few regulators, most genes are probably regulated by factors that are not directly related to their function. We call this mode of regulation indirect control. As an example, bacterial genes are often regulated by 'global' transcription factors that regulate diverse and sometimes functionally unrelated genes (Martinez-Antonio and Collado-Vides, 2003). Second, bacterial regulatory systems have evolved under very different conditions than those being tested in the laboratory. If the utility of a gene's activity correlates with a functionally unrelated signal, then regulation by that signal will be selected for in the wild, but this correlation will probably not be maintained in artificial conditions. So we do not expect indirect control that evolved in the wild to be adaptive under artificial conditions. In contrast, if there is a direct regulatory link between an environmental signal and the physiological response, as with the *lac* operon, then the regulatory system can perform well outside of the conditions that it evolved under.

To test these various theories of bacterial gene regulation, we collected genome-wide mutant fitness data and gene expression data from the metal-reducing bacterium *Shewanella oneidensis* MR-1 across 15 matching conditions. We also examined large compendia of (unmatched) fitness and expression data for this bacterium. We found that 24% of genes are detrimental to fitness in some laboratory conditions, which shows that the regulation of many genes is maladaptive in the laboratory. We confirmed that the correlation between relative expression and mutant fitness is weak, as in our previous study with just four conditions (Deutschbauer *et al*, 2011). We ruled out some technical explanations for the weak correlation, such as growth phase effects on expression, subtle variations in experimental conditions, or genetic redundancy due to paralogs, and we found little evidence of anticipatory control. As evidence of indirect control, we show that many genes are expressed constitutively instead of being controlled by transcription factors, or are regulated by growth rate. Furthermore, for many genes, this regulation seems to be suboptimal and cannot be explained by standby control.

We also show that genes with closely related functions can have rather different expression patterns, which suggests that some of them are not under direct control.

To test the generality of our findings, we examined the expression and mutant fitness of biosynthetic genes in four diverse bacteria—*S. oneidensis* MR-1, *E. coli* K-12, the ethanol-producing bacterium *Zymomonas mobilis* ZM4, and the sulfate-reducing bacterium *Desulfovibrio alaskensis* G20. In *E. coli*, biosynthetic genes that were required for fitness in minimal media but not in rich media were almost all downregulated in minimal media, but in the other three bacteria, this was often not the case. We also compared fitness and expression data for *Z. mobilis* ZM4 across 18 matching conditions, and found little correlation between relative expression and mutant fitness in *Z. mobilis* ZM4. We conclude that suboptimal regulation is widespread in bacteria, at least under laboratory conditions.

## Results

### Many genes are detrimental to fitness in some condition

We collected genome-wide data on mutant fitness and mRNA abundance for *S. oneidensis* MR-1 grown in 15 matching conditions: Luria-Bertani (LB) medium, a defined minimal medium with one of eight different carbon sources added, minimal lactate medium with one of four different stresses, or anaerobic respiration of fumarate with two different electron donors. For each condition, we measured fitness using two pools of mutants that grew for 6–8 generations and we measured gene expression from wild-type cells in exponential phase (see Supplementary Figure 2 for an overview). To ensure that the growth conditions were identical, matched fitness and expression experiments were conducted at the same time. We obtained both expression and fitness data for 3247 of the 4467 protein-coding genes in the genome. Of the protein-coding genes that we do not have data for, 69% (836/1220) are essential for growth in LB, are under 300 nucleotides, or

are repetitive elements such as transposases. The mutant fitness for each gene represents the change, across 6–8 generations of growth, in the $\log_2$ abundance of strain(s) with transposons inserted within that gene. The fitness values are normalized so that wild-type would have a fitness value of about zero: negative fitness indicates that the mutant strain is sick (relative to wild type) and that the gene's activity is important for growth in that condition, while positive fitness indicates that the mutant strain has an advantage and that the gene's activity is detrimental in that condition.

A few genes are strongly detrimental to fitness during aerobic growth on lactate: strains with insertions in these genes grow better than most other strains and the gene's fitness value is above 0.75 (Figure 1A). Furthermore, strongly detrimental genes tend to be well expressed (Figure 1A). Similarly, in all 14 conditions with strongly detrimental genes, the majority of these genes are expressed above the median gene. (There are no strongly detrimental genes in LB.) We also observed a larger number of genes with milder but potentially significant detrimental activity. As described previously, we used control experiments to estimate the reliability of each fitness measurement, which we summarize using a standard normal $Z$ score (Deutschbauer *et al*, 2011). Across 15 conditions and 3247 genes, we had 1172 fitness measurements with $z > 2.5$, while we would expect just 302 such cases by chance ($P(z > 2.5) \times 3247 \times 15 \approx 302$). For comparison, we had 5034 fitness measurements of significantly sick genes ($z < -2.5$). In all conditions, the putatively detrimental genes were more highly expressed on average than other genes were (Figure 1B), and in 11 of the 15 conditions, this difference was statistically significant ($P < 0.05$, binomial test). The high expression of detrimental genes confirms the fitness data, because it is easier for a gene to exert a detrimental effect if it is highly expressed. On the other hand, it is not clear why these genes are not downregulated to eliminate their detrimental activity.

To examine this issue more broadly, we identified genes that were detrimental to fitness in a compendium of 195 fitness experiments for *S. oneidensis* MR-1 (Deutschbauer *et al*, 2011).

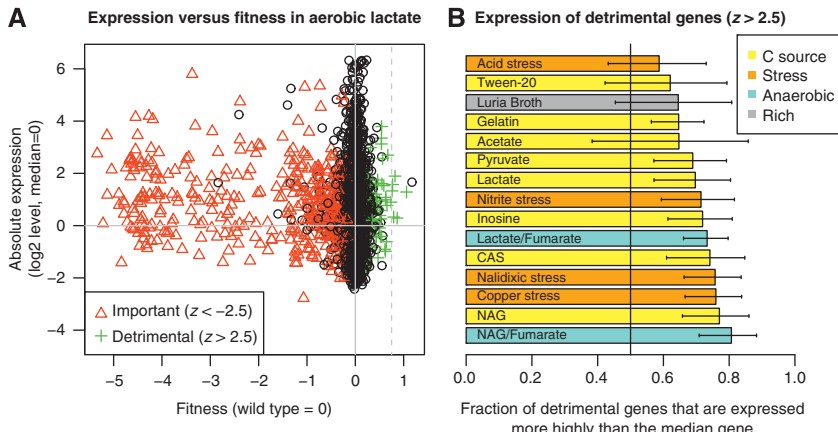

**Figure 1** In *S. oneidensis* MR-1, genes that are detrimental to fitness are highly expressed. (**A**) Absolute expression level and mutant fitness during aerobic growth in minimal lactate medium. The median gene's expression is set to 0. Genes with significant fitness effects ($|z| > 2.5$) are color-coded. The dotted vertical line at 0.75 demarcates seven strongly detrimental genes. (**B**) In all 15 conditions, genes that are detrimental to fitness ($z > 2.5$) tend to be expressed more highly than the typical gene. The vertical line shows the proportion that we would expect by chance (50%). NAG is N-acetylglucosamine and CAS is casamino acids. Error bars are 95% confidence intervals (binomial test).

Because we were interested in genes that were detrimental to growth or survival, we removed 8 experiments that measured motility, leaving us with 187 experiments. To increase sensitivity and reduce false positives, we grouped together fitness experiments that had similar patterns (pairwise correlation above 0.75), giving 38 groups. Within each group and for each gene, we required an average fitness above 0.4 as well as statistical significance from combining $Z$ scores ($P < 0.01$ after Bonferonni correction for the number of groups). In all, 798 genes (24% of the genes for which we have fitness data) were significantly detrimental in at least one group of experiments. To validate these detrimental genes, we examined adjacent pairs of genes that are cotranscribed in the same operon. Genes in the same operon often, but not always, have related functions (de Daruvar *et al*, 2002; Rogozin *et al*, 2002; Price *et al*, 2006), so if one of them is detrimental to fitness then the other should more often be detrimental. Indeed, one gene in an operon pair was much more likely to be detrimental to fitness if the other one was (53% versus 17%, $P < 10^{-15}$, Fisher's exact test). This confirms that most of these 798 genes are genuinely detrimental to fitness in some of our laboratory conditions.

These 798 genes that are detrimental to fitness are not simply selfish genes. They include a wide variety of functions, and they are not significantly depleted in any COG function category (Tatusov *et al*, 2001) or TIGR subrole (Peterson *et al*, 2001) (Fisher's exact test, false discovery rate above 0.05). Just 30 of them are annotated as potentially selfish elements such as transposases, prophages, or restriction systems. In all, 421 of the detrimental genes (53%) are important for growth or survival in another group of experiments in our compendium (fitness under $-0.4$ and $P < 0.01$ after Bonferonni correction). Some of the detrimental genes are involved in motility, which is consistent with previous reports (Langridge *et al*, 2009; Deutschbauer *et al*, 2011; Koskiniemi *et al*, 2012) and might reflect our unnaturally well-shaken growth conditions. But we doubt that motility can account for most of the detrimental genes. We previously measured mutant motility in *S. oneidensis* MR-1 by assaying the abundance of mutant strains that reached the outer ring of a soft agar plate (Deutschbauer *et al*, 2011). (These are the same experiments that were removed from the fitness compendium because they did not measure growth or survival.) In all, 34% of the 798 detrimental genes have a motility 'fitness' of under $-0.4$, as compared with 13% of other genes. Although the detrimental genes are enriched in motility genes ($P < 10^{-15}$, Fisher's exact test), motility and selfishness together only explain around a third of the detrimental genes. The regulation of the 421 genes that are sometimes detrimental and sometimes important for growth—13% of the genes that we have fitness data for—is suboptimal, at least in our laboratory conditions, as these genes 'should' be repressed when they are detrimental to growth.

To test whether genes tend to be downregulated when they are detrimental to fitness, we considered genes that are likely to be detrimental to fitness in 1 of our 15 matched conditions ($z > 2.5$) and compared their expression in that condition to their median expression across all the conditions. We excluded prophages and other potentially selfish genes from this analysis because we observed the induction of 100 prophage genes by over four-fold in nalidixic acid, which is probably a 'selfish' response to DNA damage (Qiu *et al*, 2004). We did not find a significant tendency for genes to be downregulated when they are detrimental to fitness: the mean relative expression was 0.03 for detrimental genes and 0.06 for other genes ($P < 0.1$, $t$-test; $n = 1094$ and 45 466, respectively). This shows that the detrimental activity of most of these genes cannot be explained by optimal standby control: under this model, if genes are expressed because they might be needed after a change in conditions, then they should still be downregulated (Supplementary Figure 1).

## Relative expression is little correlated with fitness

To test if bacterial gene regulation is adaptive genome-wide, we asked if genes are upregulated when they are needed for fitness and downregulated when not needed for fitness. We first compared differential expression and the difference of mutant fitness between pairs of conditions. We performed 14 comparisons derived from our 15 conditions, with aerobic growth in minimal lactate media as the common control. For example, Figure 2A shows a comparison of relative expression and differential fitness for aerobic growth in acetate versus lactate. If there was a strong relationship between relative expression and fitness, then genes that are more important for fitness on acetate than on lactate (i.e., a fitness difference below zero) would also be upregulated (i.e., an expression $\log_2$ ratio above zero), and there would be a strong negative correlation between differential fitness and relative expression. Instead, the correlation is statistically significant but is very weak ($r = -0.15$, $P < 10^{-15}$; Figure 2A). In all of the comparisons, the correlation between differential expresion and fitness is weak ($r = -0.15$ to $+0.11$).

We noticed that biosynthetic genes tend to be expressed at lower levels on acetate than on lactate, while being important for fitness in both conditions. Specifically, 67 putative biosynthetic genes (Peterson *et al*, 2001) were important for fitness in lactate and acetate (both fitness under $-0.75$) but not in LB (fitness above $-0.4$), and the average $\log_2$ levels of these genes were 1.1 in lactate and $-0.2$ in acetate ($P < 10^{-12}$, paired $t$-test). Low expression of these genes on acetate might reflect the lower growth rate of *S. oneidensis* MR-1 on acetate relative to lactate, which implies a lower flux through biosynthetic pathways. After removing biosynthetic genes, the correlation between differential fitness and relative expression is still very weak ($r = -0.08$). This anecdote illustrates that genes that are important for fitness in both conditions might change expression because of varying flux, so we focused on genes that are important for fitness in one condition but not the other or on genes that are not important for fitness in either condition.

In most of the comparisons, genes that are important for fitness in just one of the two conditions do tend to change expression in the expected direction (e.g., Figure 2B). However, over a third of these differentially fit genes change expression the 'wrong' way (i.e., lower expression on acetate for genes that are important only on acetate or lower expression on lactate for genes that are important only on lactate). The two distributions of expression changes

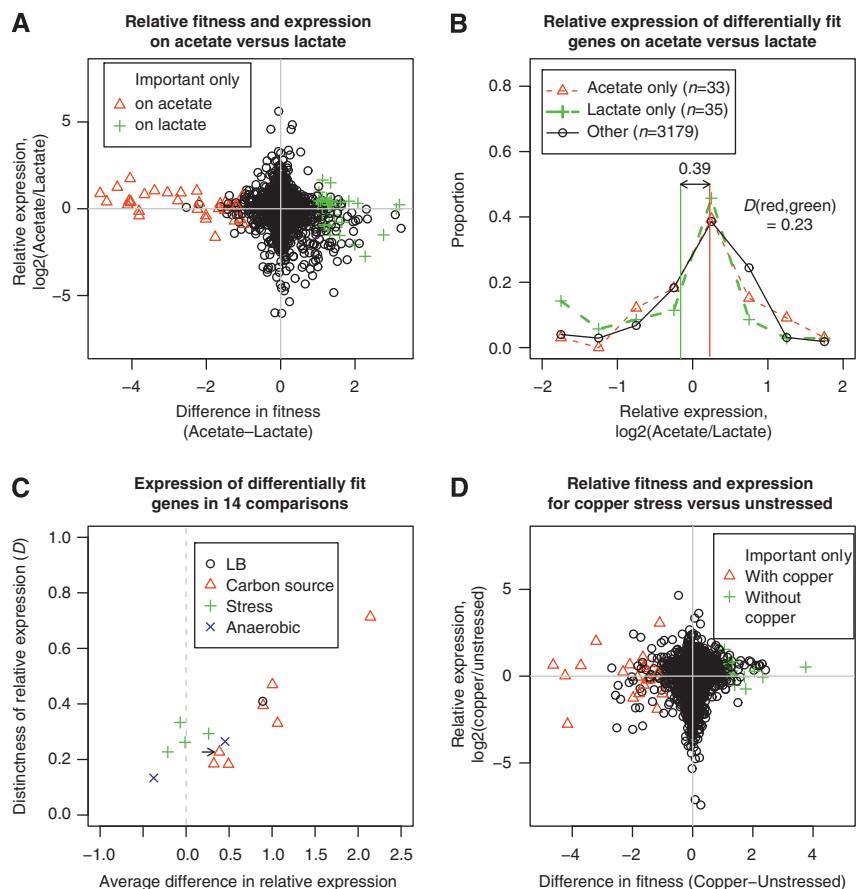

**Figure 2** In *S. oneidensis* MR-1, differential fitness and relative expression are poorly correlated. (**A**) Relative expression versus the difference in fitness for aerobic growth on acetate versus aerobic growth on lactate. Genes are color-coded if they are important for fitness on acetate or lactate but not the other condition (specifically, if fitness is below − 0.75 in that condition but not in the other condition and if the difference in fitness between the conditions is at least 1.0). (**B**) Another view of the relative expression from (A): the distribution of relative expression for genes that are only important on acetate, only important on lactate, or other genes. Out-of-range values are included in the left- or right-most bins. The vertical lines show the averages for genes that are important only in acetate (in red) or only in lactate (in green). The average upregulation of these two types of genes differs by 0.39 and the distributions overlap considerably ($D = 0.23$). (**C**) The change in expression of differentially fit genes in each of 14 conditions when compared with aerobic lactate. Each comparison is performed as in (B): the *x* axis shows the difference between the two averages and the *y* axis shows the Kolmogorov–Smirnov *D* statistic for how distinct the two distributions are. The arrow highlights the comparison between acetate and lactate from (B). (**D**) Relative expression versus the difference in fitness for cells growing in minimal lactate medium with or without copper added.

(for the two types of differentially fit genes) overlap considerably, which can be quantified with the Kolmogorov *D* statistic, which depends only on the relative ranks of the values and ranges from 0 for identical distributions to 1 for distributions that do not overlap. For acetate versus lactate, $D = 0.23$. For all of our comparisons, there is considerable overlap in the distributions of relative expression between genes that are sick only in one condition or only in the other (Figure 2C, $D = 0.12$–$0.71$). In a few conditions, genes that are differentially important for fitness are just as likely to change expression in the 'wrong' direction. For example, of 22 genes that are important for fitness with copper stress but not without it, 12 are downregulated on copper stress (Figure 2D).

To test the cases where a gene's expression changes in the opposite direction than expected given the fitness data, we examined the expression of adjacent genes that are in the same operon. For the comparison between acetate and lactate, there are 18 operon pairs in which one or both genes are expressed more highly in one condition but are important for fitness only in the other condition. For 12 pairs, the 2 genes showed the

same direction of change and for the other 6 pairs, both genes show little change in expression (both absolute $\log_2$ ratios were under 0.5). Similarly, we tested operon pairs that include genes that are important for fitness with copper stress but not without it and are downregulated during copper stress. For six of seven operon pairs, both genes were downregulated during copper stress, and for the remaining pair, the expression of both genes was little changed on copper stress (both absolute $\log_2$ ratios were under 0.25). These findings confirm the non-adaptive regulation of these genes.

Conversely, many of the genes with large changes in expression are not important for fitness in either condition. In the comparison of acetate and lactate, of 114 genes that changed expression by four-fold or more, 70 (61%) have little effect on fitness in either condition (both fitness values between − 0.4 and 0.4). For all of our comparisons, this proportion was at least 60%, with a maximum of 87% for acid stress. To test the changes in expression for the genes that are not important for fitness, we again examined the expression of adjacent genes in operons. In 81% of cases (2309 of 2835),

the other gene was upregulated or downregulated in the same direction and with an absolute $\log_2$ ratio of at least 0.5. By chance, we would expect this to occur only 21% of the time ($P < 10^{-15}$, $\chi^2$ test of proportions). (The expectation is 21% because across our 14 comparisons, expression changes by 0.5 or more in 42% of cases, and the change will be in the correct direction in half of those cases).

It is possible that the change in expression of these genes is adaptive because these genes have subtle fitness benefits in one condition but not the other. To test this, we examined genes without strong phenotypes (fitness between $-0.4$ and 0.4) and compared the genes that were upregulated by two-fold or more (relative to the median across our experiments) with genes that were downregulated by any amount. Once again we excluded prophages and other potentially selfish genes. The upregulated genes had slightly lower average fitness than the downregulated genes ($-0.01$ and $+0.001$, respectively; $P = 0.0002$, $t$-test), but the distributions were quite similar ($D = 0.05$). The two-fold upregulated genes were about as likely to be significantly sick as the downregulated genes: at a cutoff of $z < -2.5$, which corresponds to a false

discovery rate of 32% for these genes with mild phenotypes, 2.2% of upregulated genes and 1.9% of downregulated genes were significantly sick. (These proportions are not significantly different: $P > 0.2$, Fisher's exact test). Because it is difficult to measure very small differences in fitness, we cannot rule out the possibility of subtle fitness benefits of the upregulated genes. However, because the upregulated genes have very similar phenotypes as the downregulated genes, and because genes with stronger phenotypes show a modest correlation between relative expression and fitness, we suspect that the upregulation of most of the genes without strong fitness benefits is not adaptive.

Another way to ask if gene regulation is adaptive is to look at the correlation, for any given gene, between expression level and mutant fitness across the 15 conditions (see examples in Figure 3A). If a gene is more highly expressed when it is important for fitness, then we should see a strong negative correlation (e.g., *tyrA* in Figure 3A). Instead, the distribution of fitness–expression correlations for all genes is about the same as if we shuffle the data and compare a gene's fitness pattern with another random gene's expression pattern (Figure 3B).

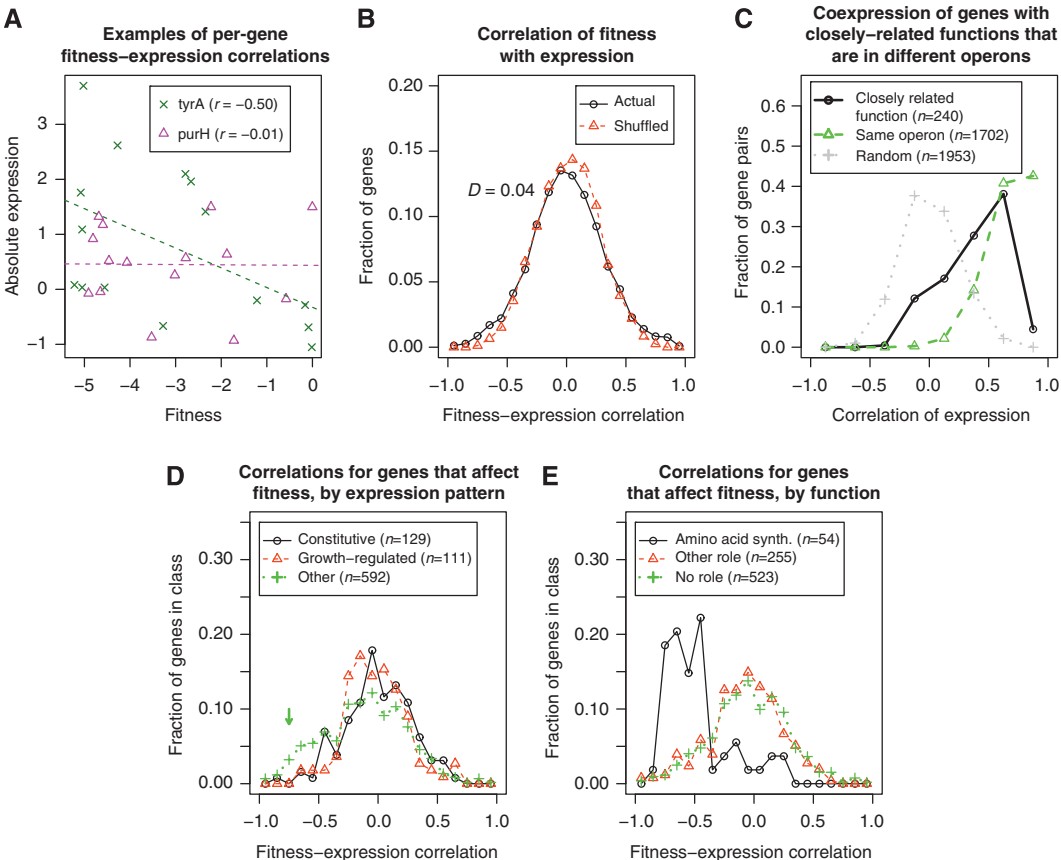

**Figure 3** In *S. oneidensis* MR-1, few genes are under adaptive control. (**A**) Absolute expression versus fitness for *tyrA* and *purH* across 15 growth conditions. The lines show the best fit for each gene: *tyrA* tends to be expressed more highly when it is more important for fitness ($r = -0.50$), but *purH* does not ($r = -0.01$). (**B**) The distribution of fitness–expression correlations, computed as in (A), for 3247 genes and for 3247 shuffled controls. (**C**) The distribution of coexpression, across 329 experiments, of pairs of genes that are not in the same operon and have closely related functions (i.e., matching TIGR subroles and similar patterns of mutant fitness across 195 experiments). We also show the distribution of coexpression for genes that are predicted to be in the same operon, as a positive control, and for random pairs of genes that have different TIGR subroles and are not adjacent or predicted to be in the same operon, as a negative control. (**D**, **E**) The distribution of fitness–expression correlations (as in B) when considering only genes that have fitness of above 0.75 or below $-0.75$ in at least one of the 15 conditions. In (D), we separate out constitutive and growth-regulated genes from other genes, and the green arrow highlights the adaptive regulation of some of the other genes. In (E), the genes are classified by their TIGR roles, which highlights the adaptive control of amino-acid synthesis genes but not other genes.

The actual distribution is significantly different from the shuffled distribution ($P = 0.01$, Kolmogorov–Smirnov test) but the difference is slight, with average correlations of 0.00 and 0.01, respectively.

We used the fitness–expression correlation of each gene to test whether genetic redundancy might be an explanation for why gene regulation does not appear adaptive. For example, if there are two partially redundant genes whose activity is important for fitness in a condition, one of them might be upregulated in that condition, but knocking it out might have only a subtle phenotype because the other gene is still active. Such redundancy is often associated with paralogs (although paralogs in *S. oneidensis* MR-1 often have detectable phenotypes when mutated; Deutschbauer *et al*, 2011). We compared the distribution of expression–fitness correlations for 392 genes in our data set that have paralogs (above 30% identity) with the distribution for genes that lack paralogs and found little difference (means of 0.014 versus 0.002, $P = 0.4$, *t*-test). Thus, genetic redundancy between paralogs does not explain the lack of a correlation between relative expression and mutant fitness in *S. oneidensis* MR-1. Also, although we did not test genetic redundancy more broadly, genetic redundancy cannot explain why genes are often detrimental to fitness.

Overall, when we compare relative expression with differential fitness, either by selecting pairs of conditions or by examining each gene across all 15 matching conditions, we find that they are weakly correlated. This strongly suggests that the regulation of many genes is not adaptive under our laboratory conditions.

## Genes with close functional relationships are often not coregulated

To confirm that gene expression patterns are often not correlated with a gene's function, we examined the coexpression of genes that have closely related functions but are not in the same operon. Using a compendium of 195 diverse fitness experiments for *S. oneidensis* MR-1 (Deutschbauer *et al*, 2011), we identified 240 pairs of genes that were highly cofit (correlation of fitness above 0.8), were annotated with the same TIGR subrole (Peterson *et al*, 2001), did not belong to the same predicted operon (Price *et al*, 2005; Dehal *et al*, 2009), and were not nearby each other in the genome (not within 10 genes of each other). When we examined the coexpression of these functionally related pairs across 329 expression experiments for *S. oneidensis* MR-1, we found that they have only a moderate tendency to be coexpressed (Figure 3C). For example, 83% of operon pairs have a coexpression of 0.5 or higher, but just 43% of the 240 functionally related non-operon pairs do. Furthermore, according to gene regulation that was predicted via comparative genomics and manually compiled in RegPrecise (Novichkov *et al*, 2010), these functionally related pairs are usually not coregulated: of the 240 pairs, there is a regulatory prediction for at least 1 gene among 97 pairs, and both genes are predicted to be regulated by the same transcription factor in only 7 cases.

To test this more carefully, we manually examined the 76 pairs of genes with a close functional relationship but little coexpression ($r < 0.3$). In all, 36 of the 76 pairs had known

functional differences or showed differences in fitness in a few conditions that might explain their limited coexpression (Data set 2). For example, genes for both proline and arginine synthesis have the TIGR subrole 'Amino acid biosynthesis: Glutamate family' and show similar fitness in most, but not all, of our conditions. It is not surprising that they might be regulated differently. These pairs reflect the limited resolution of the functional classification. Another 18 pairs of genes were from flagellar operons *fliKLMNOPQR-flhB*, *flgL1-flaG-fliD-SO_3234-fliS*, *flgFGHIJ-SO_3239.3-SO_3239.2-flgL2-flgL3*, *flgBCDE*, and *flgAMN*. Some of the differences in expression of these genes might reflect the sequential activation of different stages of assembly of the polar flagellum, which has been studied in detail in related bacteria (e.g., Prouty *et al*, 2001; Dasgupta *et al*, 2003). In *Pseudomonas aeruginosa*, 7 of these 18 pairs of genes are co-regulated and are in the same 'class' of transcripts (Dasgupta *et al*, 2003), so it is not clear that these genes are needed at different times. The remaining 22 pairs of genes were from operons with closely related functions for which there was no apparent reason for the expression to differ. Specifically, these pairs of genes were from aromatic amino-acid synthesis operons *aroA*, *aroC*, *aroE*, *aroQ*, and *aroKB*; menaquinone synthesis operons *menA*, *menB*, *menF*, and *menDHCE*; branched-chain amino-acid synthesis operons *ilvGMDA*, *ilvC*, and *ilvE*; pyrimidine synthesis genes *pyrC*, *pyrD*, and *pyrF*; methionine synthesis operons *metBL* and *metC*; lipid A synthesis genes *lpxL* and *lpxM*; mismatch repair genes *mutL* and *mutS*; and chromosome separation genes *xerC* and *xerD*. Among these genes, *mutL*, *pyrD*, *pyrF*, and *xerC* are in operons with functionally unrelated genes, while the other genes listed individually are transcribed separately, as determined using high-resolution 'tiling' microarrays and 5′-end RNA sequencing (see Materials and methods).

If gene regulation evolves to an optimum, then it is difficult to explain why the regulation of these functionally related genes or operons would be different, especially for genes that are not cotranscribed in operons with functionally unrelated genes. One possibility is that for pathways that have a low cost of expression, the first and last steps of a pathway should be regulated while the middle steps should be expressed constitutively—this can be an efficient way to transcriptionally control the flux through the pathway as demand for its product changes (Wessely *et al*, 2011). However, a low cost of expression requires a low level of expression and also that the gene's activity should not be detrimental to fitness. Instead, we found that the genes in these pairs tend to be more highly expressed on average than other genes ($P < 0.002$, *t*-test, using the median expression of each gene across our 15 conditions) and that they are more likely to be detrimental to fitness than other genes ($P = 0.005$, Fisher's exact test). Overall, the lack of coexpression for these genes with closely related functions appears to be suboptimal, but it is difficult to rule out other explanations.

## Suboptimal control via constitutive or growth-rate regulation of many genes

One explanation for why there is little correlation between fitness and expression is that some genes are expressed

constitutively and are not under adaptive regulation. Using a compendium of 329 expression experiments for *S. oneidensis* MR-1, we identified 641 putative constitutive genes (17% of the genes with expression data) that have relatively constant patterns of expression. According to RegPrecise predictions (Novichkov *et al*, 2010), these genes are much less likely than other genes to be regulated by specific transcription factors or by specialized sigma factors (3.6% versus 16.1%, $P < 10^{-15}$, Fisher's exact test). This supports the idea that these constitutive genes are not subject to adaptive control.

We also hypothesized that many genes would be regulated by growth rate, because at higher growth rates, a higher proportion of cellular resources are devoted to transcription and translation (Bremer and Dennnis, 1996). By looking for genes that were coexpressed with components of the ribosome, we identified 391 genes (10% of the genes with expression data) as putatively growth regulated. We confirmed that these genes tend to be regulated by growth, via the stringent response, by examining their promoter sequences (see Materials and methods).

Constitutive and growth-regulated genes are functionally diverse, and most types of functions are represented in both sets. For constitutive genes, the only TIGR subrole that is significantly depleted is electron transport (false discovery rate under 0.05, Fisher's exact test). For growth-regulated genes, the only TIGR subrole that is significantly depleted is anion transport (false discovery rate under 0.05).

Not surprisingly, constitutive genes and growth-regulated genes do not show a correlation between fitness and expression: across our 15 matching conditions, the two groups have mean fitness–expression correlations of 0.01 and 0.00, respectively (both $P > 0.5$, *t*-test). Together these account for 21% of the genes for which we have both fitness and expression data, so constitutive or growth-regulated expression could explain the lack of adaptive control for many genes.

These genes might lack adaptive control because the benefit of regulation would be lower than the cost of making transcription factors to regulate them. In this case, expressing them when they are not important for fitness should not be costly, so they should be weakly expressed and their activity should not be detrimental to fitness. However, 49% of growth-regulated genes and 28% of constitutive genes are detrimental to fitness in some conditions. Furthermore, detrimental genes are more likely than other genes to be growth regulated or constitutive ($P = 10^{-12}$ and $P = 0.03$, respectively, Fisher's exact test). Many of the growth-regulated detrimental genes are involved in motility, which might not be detrimental under more natural conditions. After removing genes that are important for motility (i.e., motility 'fitness' < −0.4), detrimental genes are still more likely than other genes to be constitutive or growth regulated (24 versus 16%, $P < 10^{-4}$, Fisher's exact test). We did find that constitutive genes are unlikely to be highly expressed: e.g., using the median expression in our matching conditions, only 6% of constitutive genes are expressed two-fold above the median gene, while 27% of other genes are ($P < 10^{-15}$, Fisher's exact test). In all, 278 of the constitutive genes (54% of them, or 9% of the genes that we have data for) are expressed less than two-fold above the median gene in all of our matching conditions and are also not detrimental to fitness in our compendium. Constitutive

expression of these genes might be due to the high cost of regulation. In contrast, growth-regulated genes tend to be highly expressed, with a median expression in our 15 matching conditions that is roughly three-fold higher than for other genes ($P < 10^{-15}$, Wilcoxon test). Thus, we found that many of the constitutive genes and most of the growth-regulated genes have a high cost of expression, which is not consistent with the cost-of-regulation theory.

Another potential rationale for growth regulation is that these genes have consistent but subtle defects in growth. In other words, they might always be beneficial to express, but not essential. However, manual examination of our fitness compendium suggested that growth-regulated genes tend to have variable phenotypes. Consistent with this, across 187 fitness experiments, growth-regulated genes tended to have a high standard deviation of fitness, with the average of the standard deviations being 0.87 for growth-regulated genes and 0.43 for other genes ($P < 10^{-15}$, *t*-test).

Overall, we found that functionally diverse genes are expressed constitutively or are regulated by growth rate. Some of these genes are constitutively expressed at low levels without being detrimental to fitness, so that there might not be a sufficient benefit for adaptive control to evolve. But many other constitutive or growth-regulated genes have a high cost of expression and have phenotypes that vary across conditions, so their regulation appears to be suboptimal.

## Amino-acid synthesis and catabolic pathways account for most of the genes under adaptive control

To try to identify a subgroup of genes in *S. oneidensis* MR-1 that might show more correlation between fitness and expression, we considered only the 832 genes that strongly affect fitness in at least 1 of our 15 matching experiments (maximum |fitness| > 0.75). As shown in Figure 3D, among genes that affect fitness, constitutive and growth-correlated genes still show no fitness–expression correlation (both $P > 0.4$, *t*-test), but some of the other genes do (mean −0.11, $P < 10^{-13}$, *t*-test). Of the other genes that affect fitness (not including constitutive or growth-regulated genes), 16% have strong negative fitness–expression correlations of under −0.5 and are probably under adaptive control. Many of these genes are involved in amino-acid biosynthesis (Figure 3E). For example, of the 60 genes with a fitness–expression correlation under −0.5 and an annotated TIGR subrole, 31 (52%) were involved in amino-acid biosynthesis. No other functional category was enriched in genes with strong fitness–expression correlations, but 11 of these genes are involved in the catabolism of the carbon sources we used (*fadAB*, *deoC*, *gnd*, *edd*, *zwf*, *astB*, *nagABK*, and *SO_3774*). Amino-acid synthesis and catabolic genes might be regulated adaptively because the concentrations of internal metabolites provide simple indicators of whether their activity is likely to be beneficial, because their importance for fitness varies strongly across conditions, or because unnecessary expression of these genes is particularly deleterious.

We also considered the hypothesis that the regulation of genes that are more highly expressed would be under stronger selection and hence that highly expressed genes would be

more adaptively regulated. Genes that are more highly expressed tend to have a stronger (more negative) expression–fitness correlation, but the effect is weak (Spearman rank correlation $= -0.11$, $P < 10^{-9}$). We then considered only the 'well-expressed' genes that have a phenotype in at least one of our matched conditions and which do not affect motility. More precisely, we considered genes that have a median expression, across our 15 matched conditions, of at least two-fold above the median gene. Then, we removed genes that have fitness between $-0.75$ and $+0.75$ in all of our matched conditions or have motility 'fitness' under $-0.4$. Of the remaining 76 genes, 35 are biosynthetic genes that are important for fitness in minimal media, and the median expression–fitness correlation of these biosynthetic genes is $-0.49$. For the remaining well-expressed genes, the median expression–fitness correlation is just $-0.08$, which is significantly weaker than for the well-expressed biosynthetic genes ($P < 0.001$, Wilcoxon rank sum test) and is about the same as for the less-expressed genes that have phenotypes (median $-0.07$; $P > 0.5$, Wilcoxon test). Overall, high expression does not seem to be a strong indicator of whether a gene's regulation will be adaptive in the laboratory.

## Little evidence for anticipatory control

Another possible explanation for the weak correlation between expression and fitness is that the bacterium is anticipating growth in a different environment (Tagkopoulos *et al*, 2008; Mitchell *et al*, 2009). We systematically looked for evidence of anticipatory control by considering all pairs of our conditions. Given conditions A and B, if the organism uses A to anticipate B, then genes that are required for growth on B but not on A should be upregulated on A (relative to a control condition) as compared with genes that are not required for growth in either condition. We used the median expression across the 15 conditions as the control and tested the 203 pairs of conditions that have at least 10 differentially fit genes. We found only two cases of potential anticipation that were statistically significant ($P < 0.01$, Wilcoxon test with Bonferonni correction).

The most significant effect was that growth on CAS, a mixture of amino acids, 'anticipated' growth on gelatin (corrected $P < 10^{-8}$). Rather than being a form of anticipatory control, we suspect that *S. oneidensis* MR-1 cannot distinguish growth on the peptides in gelatin from growth on amino acids, so it expresses genes for taking up peptides whenever amino acids are present. Of the 15 genes that were sick on gelatin but not on CAS and that were upregulated two-fold or more on CAS, three are involved in peptide uptake (SO_1822, SO_3194.1, and SO_3195). These may be examples of indirect control.

The other significant effect was that aerobic growth on pyruvate anticipated anaerobic growth on N-acetylglucosamine (NAG) with fumarate as the electron acceptor (corrected $P < 10^{-6}$). Of 33 genes that are important for fitness with NAG/fumarate but not on pyruvate, 7 genes were upregulated by 1.5-fold or more on pyruvate. Three of these genes form a hydrogenase operon (SO_2099:SO_2097) that is predicted to be regulated by Crp and Fnr (Novichkov *et al*, 2010), and three

of the other four genes are predicted to be regulated by Crp or Fnr (*ccmC*, *ccmA*, and *ccmH*). Both Crp and Fnr are regulators of anaerobic respiration in this organism (Saffarini *et al*, 2003; Cruz-García *et al*, 2011), and both the Crp and Fnr regulons are upregulated on pyruvate (both $P < 10^{-8}$, *t*-test) so we speculate that oxygen levels might drop during batch aerobic growth on pyruvate. Alternatively, there may be another signal for these regulators.

Broadly, we found little evidence of anticipatory control in *S. oneidensis* MR-1 across our 15 conditions. A theoretical analysis of anticipatory control suggests that, under a wide range of parameters, optimal anticipation involves a small response (relative to the response when the anticipated condition actually occurs) (Mitchell and Pilpel, 2011). So our results should not be seen as evidence that anticipatory control is not occuring; rather, they suggest that anticipatory control does not strongly affect genome-wide expression patterns and cannot explain why we observe little correlation genome-wide between mutant fitness and relative expression.

## Variation in expression during the growth phase does not explain the lack of correlation with fitness

Another potential reason for low agreement between relative expression and mutant fitness is that we measured expression at one time during the growth curve (in mid-exponential phase), while our fitness data reflect the importance of the gene throughout the growth curve. For example, if a gene is important for the early adjustment to growth in a new condition but not afterwards, then at the end of the experiment, the mutant strains would have reduced abundance and the gene's fitness would be negative, yet it would be adaptive for the gene to be less expressed in mid-exponential phase. In a previous study, we examined growth curves for 48 *S. oneidensis* MR-1 mutants with a variety of fitness values (Deutschbauer *et al*, 2011). Just two mutants grew at a normal rate but with a long lag, and most fitness defects were reflected in the growth rate during mid-exponential phase. Because most genes that affect fitness are important for growth during exponential phase when we collected samples for gene expression, growth phase effects are unlikely to explain why there is little correlation between expression and fitness.

To more directly test how the relationship between expression and fitness might vary with the growth phase, we measured expression at various points in time during batch growth in rich media (LB) or in defined medium with lactate or NAG as the carbon source. The correlation between differential expression and fitness (computed as in Figure 2A) varied across time points, but was never dramatically tighter than in our original experiments. For lactate versus LB, the original correlation was $-0.11$ and the best correlation during the time course was $-0.25$; for lactate versus NAG, the original correlation was $-0.06$ and the best was $-0.11$; and for NAG versus LB, the original correlation was $-0.25$ and the best (during the time course) was $-0.21$. The correlation between differential expression and fitness also remained moderate if we used the maximum expression of each gene during each time course. (The correlations were $-0.18$ for

lactate versus LB, $-0.06$ for lactate versus NAG, and $-0.14$ for NAG versus LB, respectively.) Thus, the time at which we measured expression does not explain the low correlation between differential expression and fitness.

## Repression of biosynthetic pathways in rich media is not the norm

To extend our analysis to diverse bacteria, we compared the expression and fitness of biosynthetic genes between rich and minimal media in four organisms: *E. coli* K-12, *S. oneidensis* MR-1, the ethanol-producing bacterium *Zymomonas mobilis* ZM4, and the anaerobic sulfate-reducing bacterium *D. alaskensis* G20. As shown in Figure 4, auxotrophic genes—genes that are annotated in biosynthetic pathways (Peterson *et al*, 2001) and are important for fitness in minimal media but not in rich media—tend to be upregulated on minimal media in *E. coli* K-12 and in *S. oneidensis* MR-1, with average log$_2$ ratios of 1.5 and 0.84, respectively ($P < 10^{-15}$ and $P < 0.001$, *t*-test). However, in *Z. mobilis* ZM4 and in *D. alaskensis* G20, auxotrophic genes are not upregulated in minimal media (both $P > 0.3$, *t*-test).

Surprisingly, in *S. oneidensis* MR-1, 28 of the auxotrophic genes are downregulated in minimal media, and 15 of these are involved in nucleotide synthesis. These genes are scattered across 11 different operons—*guaBA*, *purC*, *purEK*, *purF*, *purHD*, *purL*, *purMN*, *pyrC*, *pyrD*, *pyrE*, and *pyrF*—so this pattern has evolved independently many times. *pyrD* and *pyrE* are in operons with functionally unrelated genes, but there is no obvious reason why the other nine operons are not regulated by nucleotide availability. The expression time courses for LB, lactate, and NAG confirm that the 15 nucleotide synthesis genes are more highly expressed during log phase growth in LB—which contains nucleotides—than at any phase of growth in defined media. Although mutants in guaBA do show a mild growth defect in LB, which suggests that their activity might be required, mutants in the other nucleotide synthesis genes do not. Thus, in *S. oneidensis* MR-1, the expression of nucleotide synthesis genes does not respond to the availability of nucleotides or the cell's requirements for these genes.

We propose that *E. coli* K-12 has evolved direct regulation of biosynthetic pathways by the relevant end products so that it can efficiently utilize many different carbon sources, including amino acids and nucleotides. In particular, the switch between degrading and synthesizing these compounds may require regulation to avoid futile cycles in metabolism. In contrast, *S. oneidensis* MR-1 is adapted for utilizing amino acids but not

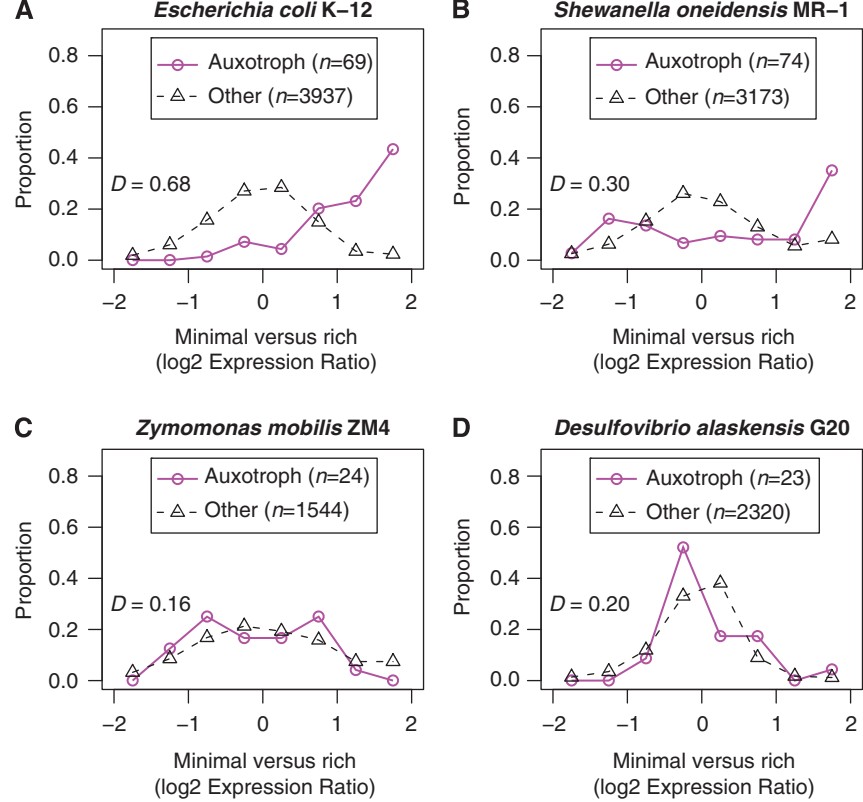

**Figure 4** Biosynthetic pathways are upregulated in minimal media in some bacteria but not in others. We examined whether auxotrophs were upregulated in minimal media, as compared with other genes, in (**A**) *E. coli* K-12; (**B**) *S. oneidensis* MR-1; (**C**) *Z. mobilis* ZM4; and (**D**) *D. alaskensis* G20. In all four organisms, the auxotrophs are annotated by TIGR role as being involved in amino acid, nucleotide, or cofactor synthesis, and experimental data confirm that they are important for growth in a defined medium but not in rich medium. For *E. coli* K-12, we used growth data of deletion mutants from the Keio collection (Baba *et al*, 2006) and expression data from Allen *et al* (2003). For the other organisms, we collected fitness data using pooled transposon mutants and we collected gene expression data using microarrays. Genes were considered as important only in defined medium if their fitness was below $-0.75$ in defined medium but not in rich medium and the difference in fitness was at least 1. The expression log$_2$ ratios are normalized so that the median value is 0. Log$_2$ ratios that are below $-2$ or above 2 are included in the left- or right-most bins, respectively.

nucleotides: it does grow on DNA or on a few nucleosides as carbon sources, but more slowly than on peptides, and it cannot utilize nucleobases (Serres and Riley, 2006; Pinchuk *et al*, 2008). A genome-scale metabolic model suggests that during growth on adenosine, deoxyadenosine, or inosine, it degrades the ribose or deoxyribose portion and secretes the nucleobases (Pinchuk *et al*, 2010). If *S. oneidensis* MR-1 is not adapted to utilizing nucleobases, then this might explain why it does not control the expression of these synthesis pathways by nucleotide availability. Finally, *Z. mobilis* ZM4 and *D. alaskensis* G20 do not, as far as we know, use amino acids or nucleotides as carbon sources and may not have encountered high levels of amino acids or nucleotides often enough for transcriptional regulation of these pathways in response to those compounds to be selected for. Overall, we found that biosynthetic pathways are often not downregulated when their end products are available.

### Little correlation between relative expression and fitness in *Z. mobilis* ZM4

To test the relationship between relative expression and fitness in another bacterium in diverse conditions, we collected mutant fitness data and gene expression data for *Z. mobilis* ZM4 across 18 conditions. As *Z. mobilis* ZM4 can only use a few sugars as carbon sources, we studied growth in rich and minimal media and in various stresses. First, we examined relative expression and differential fitness between pairs of conditions, with growth in rich media as the common control condition. Across 17 comparisons, the median correlation between relative expression and differential fitness was just $-0.01$, so there was little tendency for genes that were more important for fitness to be upregulated. (The only condition with a correlation under $-0.1$ was ethanol stress, with a correlation of $-0.22$.) Second, unlike in *S. oneidensis* MR-1, in *Z. mobilis* ZM4 there was no significant difference between the distribution of per-gene fitness–expression correlations and the shuffled distribution ($P > 0.5$, Kolmogorov–Smirnov test with 1568 genes and 1568 controls). The mean correlations were 0.007 and 0.006, respectively. After removing genes without fitness effects, constitutively expressed genes, and growth-regulated genes, the mean correlation remained at 0.007. Overall, the correlation between expression and fitness was weaker in *Z. mobilis* ZM4 than in *S. oneidensis* MR-1, which might reflect the rather artificial conditions we used, less careful matching of the experimental conditions for the two assays, or a simpler regulatory system—*Z. mobilis* ZM4 has just 65 transcription factors while *S. oneidensis* MR-1 has 243.

## Discussion

We have shown that in diverse bacteria, there is little correlation between when genes are important for fitness and when they are more highly expressed. The lack of correlation does not result from a mismatch between when we measured expression and when we measured fitness or from genetic redundancy between paralogs. In *S. oneidensis* MR-1, adaptive control seems to be rare except for amino-acid

synthesis and carbon source catabolism, and nucleotide synthesis is not under adaptive control. In *Z. mobilis* ZM4 and in *D. alaskensis* G20, few of the biosynthetic genes are under adaptive control, as their expression levels do not increase in minimal media. In contrast, in *E. coli*, most biosynthetic genes, of all types, are downregulated in rich media. Our results do not seem consistent with the traditional view that most of bacterial gene regulation is adaptive. We speculate that the traditional view is an over-generalization from the adaptive regulation of well-studied biosynthetic and catabolic pathways in *E. coli* and *Bacillus subtilis*. Instead, our results suggest that indirect control is widespread and that it leads to suboptimal expression patterns.

### Suboptimal control in the laboratory

We have shown that the misregulation of many genes is detrimental to fitness and hence is suboptimal in the laboratory. In all, 24% of genes in *S. oneidensis* MR-1 are significantly detrimental for fitness (above 0.4) in some conditions. Furthermore, detrimental genes tend to be highly expressed, and genes are not downregulated when they are detrimental (as would be expected under a model of optimal standby control). A change in $\log_2$ abundance of 0.4 across seven generations corresponds to a fitness advantage of 4% per generation ($2^{0.4/7} \approx 1.04$). This is far too large a benefit from mutating a gene to be explained by the waste of cellular resources in making unneeded protein. (Few if any proteins account for 4% of total expression.) Thus, the activity of many bacterial proteins imposes significant fitness costs in the laboratory, even at wild-type levels of expression.

Because we measured mRNA levels and not protein levels, we cannot test whether post-transcriptional regulatory mechanisms are adaptive. However, if post-transcriptional regulation was operating optimally, then it would eliminate the detrimental activities of proteins. Furthermore, in bacteria, repressing translation often destabilizes the mRNA (Deana and Belasco, 2005), so regulation of translation would affect the mRNA levels that we measured. Finally, in *E. coli*, genes with high mRNA expression tend to have high protein expression (Lu *et al*, 2007; Taniguchi *et al*, 2010), which implies a significant cost of unnecessary expression even if the protein is inactive. Thus, post-transcriptional regulation cannot explain why much of transcriptional regulation appears to be suboptimal.

In the laboratory, suboptimal control seems to be more common than adaptive control (Figure 5). Among the genes from *S. oneidensis* MR-1 that we have data for, about 8% are constitutively lowly expressed, are not detrimental to fitness, and do not have a strong correlation between mutant fitness and relative expression. These genes might lack adaptive control because the cost of regulation would not be worth it. Another 8% of genes are detrimental to fitness but are important for motility, which is probably an adaptive lifestyle in the wild but not in the laboratory. Another 1% of genes are detrimental to fitness and are potentially selfish elements such as prophages or transposons—'selfish' regulation of these genes may benefit the genes and not the host. Together, these three explanations account for just 17% of genes that we have data for. Another 5% of genes have strong fitness–expression

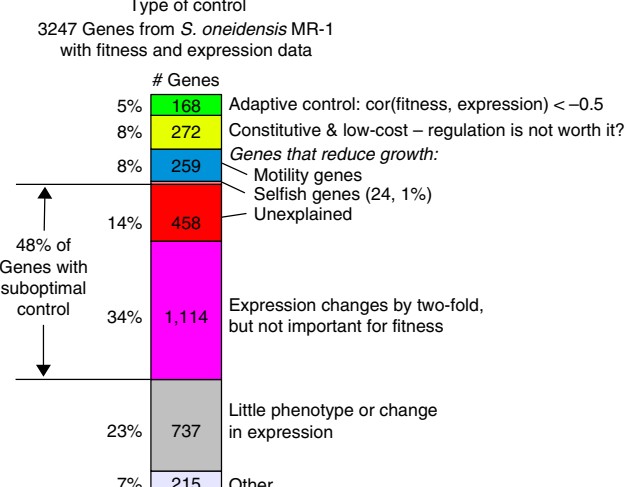

**Figure 5** Adaptive, low-cost, or suboptimal control of genes in *Shewanella oneidensis* MR-1. Among the genes with both fitness and expression data, we classified their control by the following criteria. If a gene fit into multiple categories, then it was counted only in the first (top-most) category. First, we classified genes as being under adaptive control if the fitness–expression correlation, across 15 matched conditions, was under − 0.5. We used a threshold of − 0.5 because this is roughly where the actual distribution of fitness–expression correlations diverges from the shuffled distribution (Figure 3B); also, 53% of amino-acid synthesis genes are below this threshold. We classified genes as constitutive and low cost if they had a low standard deviation of expression (in a large compendium), they were not detrimental to growth (in 38 groups of fitness experiments), and their absolute expression level was at most two-fold above the median gene in all of our 15 conditions. Genes that are significantly detrimental to growth in 1 or more of 38 groups of fitness experiments were subclassified into genes that are important for motility (motility 'fitness' below − 0.4), selfish genes such as transposons, prophages, and restriction elements, or other unexplained genes. Genes were considered to change expression without being important for fitness if, in any of 14 comparisons between conditions, the expression changed by two-fold or more but the fitness value was between − 0.4 and 0.4 in both conditions. The remaining genes were classified as having little phenotype or change in expression if their fitness value was between − 0.75 and + 0.75 in all 15 matched conditions.

correlations and are probably under adaptive control. In contrast, 48% of genes are under suboptimal control, at least in our laboratory conditions: they are either detrimental to fitness, without being explained by motility or selfishness, or they are strongly upregulated or downregulated between conditions without being important for fitness in either condition (Figure 5). Another 23% of genes have little phenotype or change in expression in our conditions, so we cannot determine if their control is adaptive or not. The remaining 7% of genes had phenotypes in our matched conditions but their expression was neither strongly adaptive nor strongly suboptimal. As they had a mean fitness–expression correlation of + 0.01 (which is not significantly different from zero, $P > 0.4$, *t*-test), we suspect that the regulation of many of these genes is suboptimal as well.

## Suboptimal control in the wild

According to our model of indirect control, gene expression responses will be more adaptive if examined under natural conditions than in the laboratory. Intuitively, we are confusing the bacteria by growing them in unfamiliar conditions such as high nutrient levels, high cell densities, pure carbon sources, no competition from other microorganisms, and no predation. Also, indirect control may have evolved because of correlations between environmental parameters that occur in the wild but not in our laboratory experiments. Measuring gene expression during slow growth at low cell densities in the presence of other microorganisms seems challenging. Nevertheless, given the rapid rate of improvements in DNA and RNA sequencing, we hope that it will soon become feasible.

Although we predict that bacterial regulation will perform better under natural conditions, several features of bacterial gene regulation seem likely to be suboptimal in the wild as well. First, we found many cases where genes with closely related functions had rather different expression patterns. Although this appears suboptimal, for pathways with a low cost of expression, it can be optimal for some steps to be constitutive and some steps to be regulated (Wessely *et al*, 2011). Because the genes in our cases tended to have a high cost of expression, this theory does not seem to apply, and we believe that the regulation of these genes is suboptimal. However, there could be other explanations that we have not considered. Second, many operons contain functionally unrelated genes, which seems suboptimal (de Daruvar *et al*, 2002; Rogozin *et al*, 2002; Price *et al*, 2006). In the stomach bacterium *Helicobacter pylori*, operons consist predominantly of functionally unrelated genes (Price *et al*, 2005; Sharma *et al*, 2010). Third, although operons tend to be conserved across related bacteria (Wolf *et al*, 2001; Ermolaeva *et al*, 2001), operons are rarely conserved between distantly related bacteria, even if they contain functionally related genes (Itoh *et al*, 1999). When operon structures change, gene expression patterns change as well, so it seems unlikely that gene regulation is optimal both before and after the change (Price *et al*, 2006). Fourth, theoretical analysis of the transcriptional regulation of biosynthetic pathways suggests that the optimal design is for them to be regulated by their end product, but many pathways are instead regulated by transcription factors that sense metabolic intermediates (Chubukov *et al*, 2012). This seems suboptimal and is also consistent with our proposal that sensors for the optimal signals might not be available.

## Indirect control

We proposed that the low correlation between relative expression and mutant fitness reflects indirect control of most genes by factors that are unrelated to the function of the gene. We presented more evidence against alternative models than evidence for indirect control, but we do have two findings that argue for indirect control. First, many genes, with diverse functions, are expressed constitutively or are regulated by growth rate. As a class, these genes show no correlation between relative expression and mutant fitness. Second, genes with a close functional relationship often have rather different expression patterns if they are not in the same operon; thus, these genes are probably not regulated by the same signals.

We proposed that indirect control occurs partly because of the limited number of regulators present in bacterial genomes. Indirect and suboptimal control might also evolve more

rapidly than adaptive direct control. For example, specific transcription factors or specific binding sites are not required to evolve constitutive or growth-regulated control. Indirect control by global regulators may also evolve rapidly: because global regulators are present at high concentrations, they will bind at low-affinity sites that require relatively little information to specify (Sengupta *et al*, 2002; Lozada-Chávez *et al*, 2008), so these sites should evolve more readily than binding sites for other regulators (Stone and Wray, 2001; Berg *et al*, 2004).

Our theory rests on the empirical observation that bacterial genomes have far more operons than transcription factors. For example, *S. oneidensis* MR-1 has 4467 protein-coding genes and around 2800 transcription units but only 243 transcription factors (5.4% of proteins). What limits the number of transcription factors in bacterial genomes? There is a roughly linear relationship between the number of proteins encoded by a bacterial genome and the proportion of genes that encode transcription factors (van Nimwegen, 2003). The relatively small number of transcription factors in smaller bacterial genomes suggests that the benefits of additional control would be less than the costs or would be too small for selection to operate. This might reflect the adaptation of bacteria with small genomes to narrow niches. For example, we found little correlation between relative expression and fitness in *Z. mobilis* ZM4, which utilizes only three different carbon sources and has just 65 transcription factors among its 1892 protein-coding genes. In bacteria with large genomes, transcription factors are often acquired by horizontal gene transfer (Price *et al*, 2008), but the acquisition of additional transcription factors might be limited because transcription factors that have similar DNA binding preferences will interfere with each other (similar to the theory of Itzkovitz *et al*, 2006). If the acquisition of a transcription factor that senses the relevant signal is selected against, then it might take a long time for a new sensor to evolve.

## Alternative explanations for suboptimal control

Although we considered several other explanations for suboptimal control, such as standby control, anticipatory control, or weak selection on gene regulation, we do not believe that they are sufficient to account for our results. First, if genes are under standby control and are expressed when they are not important for fitness because they might be needed in the future, then they should still be somewhat downregulated when they are not useful (Supplementary Figure 1), but this is not what we found. Conversely, we found that genes are not downregulated when they are detrimental to fitness. Second, we looked for evidence that *S. oneidensis* MR-1 uses one condition to anticipate growth in another condition, but we found little evidence of it. Furthermore, anticipatory control is predicted to occur along with adaptive control and to have smaller effects on expression patterns (Mitchell and Pilpel, 2011). Third, although weak selection might explain why some of the weakly expressed genes are constitutive, we found that many genes are strongly detrimental to fitness in some conditions and that many of the other genes with apparently suboptimal expression

patterns (i.e., growth regulation and/or no correlation between expression and fitness) are highly expressed. The regulation of these genes should be under strong selection.

Another explanation for suboptimal control and a weak correlation between expression and fitness is that many promoters are poorly 'insulated' from environmental factors (Sasson *et al*, 2012). Even if genes are regulated by transcription factors that sense functionally relevant signals, their expression also fluctuates due to irrelevant differences in environmental conditions (Sasson *et al*, 2012). For example, their promoters might bind other transcription factors at weak sites that evolve neutrally and are not deleterious enough for selection to remove them (Lynch, 2007). Or the concentration of active transcription factor might fluctuate due to factors besides the signal that the transcription factor senses.

Poor insulation is like indirect control in that the gene's expression responds suboptimally to irrelevant signals, but the effect is proposed to evolve neutrally rather than in response to environmental correlations. We expect poor insulation to reduce the correlation between when a gene is important for fitness and when it is more highly expressed, but we are not sure that it can explain why most genes show no correlation at all. We also showed that constitutive expression and regulation by growth rate are widespread, which does not fit the insulation theory. Furthermore, we found that many genes can be detrimental to fitness, which implies strong selection on misregulation, which should remove the interfering sites. On the other hand, when we considered genes that have a close functional relationship but are not in the same operon, we saw more coexpression than we might expect from the slight correlation between expression and fitness for most genes (e.g., compare Figures 3C and D). This might be explained by poor insulation—if two promoters are responding to transcription factors that sense relevant signals, but the concentrations or activities of those transcriptions factors are affected by irrelevant changes in growth conditions, then expression from those promoters would be well correlated with each other yet fitness–expression correlations would be modest.

Another possible reason for the weak correlation between expression and fitness is that optimal control requires complex combinatorial regulation. Among genes with characterized regulation in *E. coli* (Gama-Castro *et al*, 2011), 962 of 1641 genes (59%) are regulated by more than one transcription factor. One possible reason for why combinatorial control is widespread is to make up for the relatively limited number of sensors. We speculate that combinatorial logic might perform poorly in laboratory conditions. For example, even if the sensed signals are functionally relevant, the way in which they are combined might be adapted to natural conditions. We also suspect that combinatorial control implies a rugged fitness landscape for selection on the promoter region, which might make it difficult for optimal control to evolve.

Overall, we have shown that the regulation of most bacterial genes is not adaptive, at least not as traditionally understood to involve responding to a physiologically relevant signal. In *S. oneidensis* MR-1, we found that almost half of genes are under suboptimal control in the laboratory, while far fewer are under adaptive control. To further understand the ecological

role of bacterial gene regulation, we will need to measure fitness and expression under more natural conditions.

## Materials and methods

### Fitness and expression data for *S. oneidensis* MR-1

We collected matching mutant fitness and gene expression data for *S. oneidensis* MR-1 (ATCC 700550) in 15 conditions: aerobic growth in LB broth; aerobic growth in defined minimal media with 8 different carbon sources (20 mM D,L-lactate, 20 mM pyruvate, 10 mM acetate, 20 mM NAG, 5 mg/ml mixed amino acids (CAS), 1 mg/ml gelatin, 0.5% Tween-20, or 7.5 mM inosine); aerobic growth in defined lactate medium with four different stresses (70 μM copper(II) chloride; 1 mM sodium nitrite; 1.5 μM nalidixic acid, an inhibitor of DNA gyrase; or acid stress at pH 6); and anaerobic growth in a defined medium with 20 mM D,L-lactate or 20 mM NAG as the carbon source and 30 mM fumarate as the electron acceptor. Our defined medium contained 30 mM PIPES buffer, salts (1.5 g/l $NH_4Cl$, 0.1 g/l KCl, 1.75 g/l NaCl, 0.61 g/l $MgCl_2$ż$6H_2O$, 0.6 g/l $NaH_2PO_4$), Wolfe's vitamins, and Wolfe's minerals, at pH 7. For the stress experiments, the carbon source was 20 mM D,L-lactate. For growth at pH 6, we used 30 mM MES buffer instead of PIPES. All *S. oneidensis* MR-1 samples were grown at 30°C with shaking at 200 r.p.m.

For each condition, we collected gene expression data from wild-type cells and fitness data from two pools of transposon mutants, and all three cultures for a given condition were initiated at the same time with the same media. Samples for gene expression were collected in exponential phase, and samples for fitness were collected after 6–8 doublings of the population. In pilot experiments, it made little difference whether we collected fitness data in late exponential phase or in stationary phase (data not shown).

For three conditions, we also measured gene expression during batch growth. We collected cells at varying times after inoculation of batch aerobic growth at $OD_{600}$ of 0.1 on minimal lactate medium (7 samples and maximum OD = 0.55), minimal NAG medium (6 samples and maximum OD = 1.6), and LB (7 samples and maximum OD = 4.0).

For fitness experiments, strain abundance was quantified using a microarray as described previously (Deutschbauer *et al*, 2011). Briefly, we extracted genomic DNA, used PCR to amplify the tags that 'barcode' each strain, hybridized the amplified tags to a Affymetrix 16K TAG4 array, and scanned the array (Pierce *et al*, 2007). Each strain's barcode actually contains two different tags—we amplified the 'uptags' from one pool and the 'downtags' from the other pool, mixed them together, and hybridized them to one array.

Fitness values for each strain were computed from the $log_2$ ratio of abundance after growth versus the start of the experiment. Fitness values for each gene were the average of the per-strain values. Because we use two pools of mutants that are grown and assayed separately, and because some strains are present in both pools, we can verify the reliability of a fitness experiment by asking whether strains gave similar values from both pools. We quantified this by looking at the correlation of these strains' fitness values across the two pools. In our typical fitness experiment for *S. oneidensis* MR-1, the correlation of strain fitness values was 0.92, and all experiments had correlations above 0.8 except for NAG/fumarate ($r = 0.66$). In the NAG/fumarate experiment, pairs of genes in the same operon did have well-correlated fitness values ($r = 0.66$, as compared with $r = 0.63$ in our typical experiment).

We believe that the phenotypes of these mutants are usually due to the loss of protein function. First, for 1646 of the genes, we have fitness data for strains with insertions at more than one location within that gene, and the fitness data for different insertions within a gene are quite consistent ($r = 0.87$–$0.97$ in the 15 experiments). Second, we previously complemented 10 of these mutants, including 7 insertions within hypothetical proteins (Deutschbauer *et al*, 2011). Third, a caveat in using mutants with transposon insertions is that the phenotype can be due to polar effects, in which the mutation in an upstream gene affects the expression of downstream genes in an operon. We previously showed that insertions within upstream genes often lack the phenotypes of insertions within downstream genes, which suggests that polarity is not a dominant factor in these pools of

mutants (Deutschbauer *et al*, 2011). Also, for studying whether the expression pattern of an operon is adaptive or not, it is not essential to know which gene in the operon is responsible for the observed phenotype.

To quantify gene expression, we used a 12-plex Nimblegen microarray in which each sector has 122 643 spots and 40 881 distinct probes as described previously (Deutschbauer *et al*, 2011). Briefly, we used RNAProtect (Qiagen), isolated total RNA (RNAeasy mini kit, Qiagen), prepared first-strand labeled cDNA (SuperScript Plus Indirect cDNA Labeling Module, Invitrogen), and hybridized the labeled cDNA to the microarray according to Nimbelegen's instructions. Within each experiment, the log level of expression of genes in the same operon was highly correlated ($r = 0.75$–$0.88$ for matching experiments, but growth curve experiments had values as low as 0.62). Furthermore, in each comparison of gene expression between aerobic growth in lactate and 1 of the other 14 matched conditions, the log ratios of genes in the same operon were highly correlated ($r = 0.80$–$0.90$).

### Compendium of expression data for *S. oneidensis* MR-1

We obtained 371 expression experiments from the MicrobesOnline web site (Dehal *et al*, 2009), derived primarily from Liu *et al* (2005); Faith *et al* (2008); and Deutschbauer *et al* (2011) and similar works. We removed experiments and genes with a high proportion of missing values, leaving data for 3844 genes across 329 experiments.

### Constitutive and growth-regulated genes in *S. oneidensis* MR-1

We classified genes as constitutive if the standard deviation of their $log_2$ expression ratios, across 329 conditions, was under 0.5. Although this threshold is somewhat arbitrary, it was validated by the finding that few of these genes are predicted to be regulated by specific factors.

To identify growth-regulated genes, we examined the expression patterns of 24 essential protein components of the ribosome (*rplBCDFJLMNORTWX* and *rpsBEHIJLMNPQS*). As expected, these genes are quite coexpressed, with a median pairwise correlation of 0.83. We used the average expression profile of these ribosomal genes to identify other putatively growth-correlated genes. Specifically, we identified 391 genes whose coexpression with the profile is above 0.5, including all of the original 24 genes. These 'growth-regulated' genes are only slightly less likely than other genes to be regulated by specific transcription factors according to RegPrecise (10.7 versus 14.4%, $P = 0.054$, Fisher's exact test). Nevertheless, we can confirm that they are growth regulated by examining their promoter sequences. In *E. coli* and presumably in *S. oneidensis* MR-1 as well, the growth regulation of ribosomal protein genes is mediated by the alarmone ppGpp and the DksA protein as part of the stringent response (Lemke *et al*, 2011). DksA binds to RNA polymerase and alters the efficiency of transcription initiation depending on various factors including the concentration of the first (initiating) nucleotide and a GC-rich 'discriminator' between the −10 box and the initiation site (Travers, 1980; Paul *et al*, 2004; Haugen *et al*, 2006). We used a combination of high-resolution 'tiling' microarrays and 5′ RNA-Seq to map the exact 5′ ends of transcripts for 1236 genes or operons from *S. oneidensis* MR-1 (see below). We found a substantial difference in the initiating nucleotides between growth-regulated and other transcripts: just 25% of growth-regulated transcripts begin with adenosine, while 51% of other transcripts do ($P < 10^{-7}$, Fisher's exact test). The putative growth-regulated promoters also have a higher GC content at positions −4 to −1 than other promoters do (68 versus 55%, $P < 10^{-5}$, t-test). Thus, many of the putative growth-regulated promoters in *S. oneidensis* MR-1 are affected by the stringent response.

### Transcript structures of *S. oneidensis* MR-1

We grew *S. oneidensis* MR-1 in minimal lactate media and collected high-resolution 'tiling' microarray data and performed RNA sequencing targeting the 5′ ends of transcripts, using protocols described previously (Price *et al*, 2011). Briefly, we extracted RNA from frozen

cell pellets using RNeasy miniprep columns with DNase treatment (Qiagen), confirmed RNA quality with Agilent bioanalyzer, and depleted ribosomal RNA with MICROBExpress kit (Ambion). For the tiling experiment, we then created labeled first-strand cDNA with SuperScript (Invitrogen) to hybridize to a microarray (Nimblegen) with 2.01 million probes of 60 nucleotides each. For the 5′ RNA-Seq experiment, we used terminator 5′-phosphate-dependent exonuclease (Epicentre) to remove degraded transcripts, converted 5′-triphosphate to 5′-monophosphate ends with RNA 5′ polyphosphatase (Epicentre), ligated adapters onto the 5′ end with T4 RNA ligase (Ambion), used random hexamer primers that also included a sequencing adaptor to create cDNA, and used PCR amplification to enrich for DNA that contained both adaptors (see Price *et al*, 2011 for details). The 5′ RNA-Seq data (Illumina) gave 21.5 million reads that mapped uniquely to the genome. To identify transcript starts, we combined local peaks in the 5′ RNA-Seq data with sharp rises in the tiling data (Price *et al*, 2011). Specifically, we used local peaks in the 5′ RNA-Seq data that had at least 50 reads and we required these starts to be within 30 nucleotides of a sharp rise in the tiling data that had a local correlation to a step function (Güell *et al*, 2009) of at least 0.8. We associated a transcript start with a gene if it was up to 200 nucleotides upstream of the 5′ end of the gene. For transcript start analyses, we considered only genes on the main chromosome.

## Fitness and expression data for *Z. mobilis* ZM4

Our standard growth condition for *Z. mobilis* ZM4 (ATCC 31821) was aerobic growth at 30°C in a rich medium with 25 g/l glucose, 10 g/l yeast extract, and 2 g/l $KH_2PO_4$. We collected fitness and expression data for *Z. mobilis* ZM4 grown in this condition and with various inhibitory compounds added, namely 0.45% furfuryl alcohol, 4 mM 4-hydroxybenzaldehyde, 5–10 mM 3-hydroxybenzoic acid, 7% ethanol, 0.09–0.12% acetic acid, 0.2% acetic acid, 7.5 mM 5-hydroxymethyl-furfural, 1% butanol, 9.9–12.5 mM furoic acid, 17–26 mM levulinic acid, 0.1–0.2 M NaCl, 3–6 mM hydroquinone, 0.0004–0.00055% hydrogen peroxide, 2.5 mM vanillin, or a complex stress provided by 6–8% hydrolyzed plant material. Some of the concentrations are given as ranges because the fitness experiments were done at more than one concentration or at a different concentration from the expression experiments. If the fitness experiments were done at more than one concentration or more than once then we averaged them. The correlation of the per-gene fitness values from experiments with different concentrations of the same inhibitor was usually above 0.8, with one exception. We also collected fitness and expression data for growth in rich media at 37°C and for growth at 30°C in a defined medium containing 20 g/l glucose, salts, and vitamins (Goodman *et al*, 1982). Fitness was measured using a similar approach as in *S. oneidensis* MR-1; the two pools of transposon insertions that we used will be described in more detail elsewhere (JMS *et al*, submitted). Most of the fitness experiments for *Z. mobilis* were part of this other study; the fitness experiments that are specific to this study were for 7% ethanol, 1% butanol, and growth at 37°C. In the typical experiment for *Z. mobilis* ZM4, the correlation of strain fitness values between the two pools was 0.94, and all experiments had correlations above 0.8. We measured gene expression with a microarray from Nimblegen with 51 851 probes for 1882 genes, using the same protocols as for *S. oneidensis* MR-1. Within each experiment, the log level of expression of genes in the same operon was correlated ($r = 0.58–0.82$). Also, for each experiment, the log ratio of expression between that condition and the rich media control was correlated for genes in the same operon ($r = 0.59–0.79$).

## Constitutive and growth-regulated genes in *Z. mobilis* ZM4

We considered genes in *Z. mobilis* ZM4 to be constitutively expressed if the standard deviation of their absolute expression level, across our 18 conditions, was under 0.2. This accounted for 117 genes (7% of the genes that we had both expression and fitness data for). We identified growth-regulated genes by taking the average expression profile of 48 ribosomal proteins and identifying genes that were coexpressed with

this profile ($r > 0.5$). This selected 352 genes (22% of the genes that we had both expression and fitness data for).

## Fitness and expression data for *D. alaskensis* G20

We grew *D. alaskensis* G20 (provided by Terry Hazen, University of Tennessee, Knoxville) anaerobically at 30°C in a defined lactate-sulfate medium (LS4D) and in a similar medium supplemented with yeast extract (LS4), as described previously for *D. vulgaris* Hildenborough (Price *et al*, 2011). We collected fitness data using a similar approach as in *S. oneidensis* MR-1; the two pools of transposon insertions that we used will be described in more detail elsewhere (JVK *et al*, in preparation). Unlike in *S. oneidensis* MR-1 or *Z. mobilis* ZM4, we used separate chips to assay the two pools for a given condition: for each sample, we amplified both the uptags and the downtags and we hybridized those to the same array. We averaged the $log_2$ intensities of the uptags and downtags together before further processing. In both rich and minimal media, strain fitness was highly consistent between the two pools ($r = 0.94$ and $r = 0.97$, respectively).

We measured gene expression in *D. alaskensis* G20 with a high-resolution 'tiling' microarray (Nimblegen) with 2.1 million 60-mer probes, using the same protocols as with the *S. oneidensis* MR-1 tiling array. We considered only probes for the coding strand of genes, we used quantile normalization to put the two data sets into the same distribution, and we averaged the normalized $log_2$ intensities across the probes for each gene. Genes in the same operon had highly correlated expression differences between rich and minimal medium ($r = 0.87$).

## Analysis of mutant fitness data

In previous work on fitness data from *S. oneidensis* MR-1 (Deutschbauer *et al*, 2011), we normalized the fitness values so that the median strain had a fitness of zero. Because there can be differential efficiency in extracting DNA of different sizes, we did this separately for the main chromosome and the megaplasmid. We had found that some experiments had significant effects depending on which microplate the strain was grown on during assembly of the pools, so we also normalized the data so that each 'pool plate' had a median fitness of zero. Here, we used pool-plate normalization for *S. oneidensis* MR-1 and for *Z. mobilis* ZM4, but it was not needed for *D. alaskensis* G20.

We also identified a small trend by chromosome position in some fitness experiments. Specifically, there was sometimes a correlation between fitness and the distance from the origin of DNA replication. This might result from collecting the start and end samples at different growth stages—if the cells are rapidly dividing then the area near the origin of replication will be at higher copy number. To remove this effect, for strains on the main chromosome, we computed a smooth estimate of how the fitness of each strain varied with chromosomal position (using the loess function in R) and we subtracted this from the fitness values.

It appears that the median gene in *Z. mobilis* ZM4 has a fitness defect in most conditions. For example, in all of our experiments, the median fitness of genes with annotated functions was below the median fitness of purely hypothetical proteins. This might reflect its relatively small genome
(1892 proteins). Thus, setting the median gene's fitness to zero was not appropriate. Instead, for genes on the main chromosome, we set the mode of the distribution to zero. (More precisely, we estimated the mode by finding the maximum of the kernel density, using the density function in R with default settings, and we subtracted the mode from the values.) Mode-based centering typically lowered the fitness values by around 0.1. We used mode-based centering for *S. oneidensis* MR-1 and *D. alaskensis* G20 as well, although it made less difference for those organisms.

To identify genes with strong effects on fitness, we used a threshold of $\pm 0.75$. A fitness of $\pm 0.75$ corresponds to around a 7% change in abundance per generation. Effects above this magnitude were usually statistically significant. For example, in the 15 matched experiments in *S. oneidensis* MR-1, genes with fitness effects of $\pm 0.75$ or stronger have $|z| > 2$ in 83–99% of cases (95% in the median experiment).

Fitness *z* scores were computed as described previously. Briefly, we used a *t*-like test statistic for each gene to summarize the consistency of the measurements for its strains. This statistic also takes into account how noisy the data for other genes appears to be. Then, we transformed the test statistic to fit the standard normal distribution by using 'fitness' data from control experiments in which we independently recovered the pools from the freezer and assayed their relative abundance (Deutschbauer *et al*, 2011).

To identify genes with more subtle but reproducible effects on fitness, we grouped together experiments with similar patterns (those having a pairwise correlation of above 0.75). For each group, we used Fisher's method to combine the significance of genes (as assessed using *z* scores). For each gene, we corrected for multiple testing across groups.

## Statistical tools

All statistical analyses were conducted in R 2.11 or 2.13 (http://r-project.org/). Data were visualized in R and in MicrobesOnline (Dehal *et al*, 2009).

## Data availability

All fitness data are available in MicrobesOnline (http://microbesonline.org/). Fitness data for *S. oneidensis* MR-1 are also availabe as Data set 1. All gene expression, tiling, and 5′ RNA-Seq data are available in the Gene Expression Omnibus, including expression data for *S. oneidensis* MR-1 (GSE39462), tiling data for *S. oneidensis* MR-1 (GSE39468), 5′ RNA-Seq data for *S. oneidensis* MR-1 (GSE39474), expression data for *Z. mobilis* ZM4 (GSE39466), and tiling data for *D. desulfuricans* G20 (GSE39471). All data and source code are available from the authors' web site (http://genomics.lbl.gov/supplemental/exprVfitness2012/).

## Supplementary information

## Acknowledgements

We thank Dacia Leon, Dan Tarjan, Keith K Keller, Jason K Baumohl, and Marcin P Joachimiak for technical assistance, and Paramvir S Dehal for helpful discussions. We thank the Energy Biosciences Institute for providing the mutant collection for *Z. mobilis* ZM4. This work conducted by ENIGMA was supported by the Office of Science, Office of Biological and Environmental Research, of the US Department of Energy under Contract No. DE-AC02-05CH11231. The funders had no role in study design, data collection and analysis, decision to publish, or preparation of the manuscript.

*Author contributions:* APA, AMD, and MNP conceived the project. AMD, JMS, KMW, JSM, JVK, and WS collected data. MNP and TR analyzed the results. MNP wrote the paper.

## Conflict of interest

The authors declare that they have no conflict of interest.

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
