## [Review Process File · Molecular Systems Biology]

Indirect and Suboptimal Control of Gene Expression is Widespread in Bacteria

Morgan N. Price, Adam M. Deutschbauer, Jeffrey M. Skerker, Kelly M. Wetmore, Troy Ruths, Jordan S. Mar, Jennifer V. Kuehl, Wenjun Shao, Adam P. Arkin

Corresponding author: Morgan N. Price, Lawrence Berkeley Lab

Review timeline:	Submission date:	05 March 2013
	Editorial Decision:	04 February 2013
	Revision received:	05 March 2013
	Accepted:	13 March 2013

Editor: Thomas Lemberger

Transaction Report:

1st Editorial Decision

04 February 2013

Thank you again for submitting your work to Molecular Systems Biology. First of all, I would like to apologize for the delay in getting back to you, which was due to the late arrival of the reports after the end of year holiday season. We have now finally heard back from the three referees who agreed to evaluate your manuscript. As you will see from the reports below, the referees find the topic of your study of potential interest. They raise, however, substantial concerns on your work, which should be convincingly addressed in a revision of the manuscript.

While reviewer #1 and #2 are generally supportive, reviewer #3 is less convinced about the study. One important point raised by this reviewers is whether the conclusions remains robust independently of the definition of fitness. A second point, raised also by reviewer #2, is the need to analyze the expression-fitness 'incongruence' in the light of the natural ecology of the organisms (ie ecologically relevant media vs 'non-familiar' environments).

If you feel you can satisfactorily deal with these points and those listed by the referees, you may wish to submit a revised version of your manuscript. Please attach a covering letter giving details of the way in which you have handled each of the points raised by the referees. A revised manuscript will be once again subject to review and you probably understand that we can give you no guarantee at this stage that the eventual outcome will be favorable.

REFeree REPORTS:

Reviewer #1 (Remarks to the Author):

My apologies for the late review. This is a deeply interesting and valuable paper that emphasizes the

loose genotype-phenotype mapping in bacteria. Within the framework of traditional evolutionary thinking, it is highly unexpected that genes with measurable negative fitness affect are so often not down-regulated at all.

I have rather minor comments that only relate to presentation/discussion.

1. The article would benefit from being more explicit regarding the accuracy of the expression measurements.
2. The authors do not really discuss in depth the rather scandalous question prompted by their own results: if genes are typically not upregulated when they are needed and not downregulated when their expression is detrimental, how come there is so much elaborate expression regulation in bacteria?! This discrepancy cries for explanations. One important factor that is likely to drive the selection for regulation is the balance hypothesis (e.g. Birchler JA, Veitia RA. Gene balance hypothesis: connecting issues of dosage sensitivity across biological disciplines. Proc Natl Acad Sci U S A. 2012 Sep 11;109(37):14746-53) but there are other factors as well, beyond doubt. The article surely can be improved by a more focused discussion of this issue.
3. The article suffers to some extent from quasi-teleological language (gene expression is regulated or not regulated because of these and these reasons etc). I do not suspect that the authors are actually prone to teleological thinking but modifying the text to avert this impression would be helpful.

Reviewer #2 (Remarks to the Author):

This is a very interesting paper, which brings very surprising results that are counter intuitive, contradict expectations and as such should be published. The work is performed very well, paper is very clearly written and I'm sure that the work will evoke much intensive research as a follow up. MSB would serve as an ideal host for this important work.

The main findings of this paper are:

- a. "Detrimental genes (i.e. genes that the cell does better w/o them) are typically not down-regulated in the condition in which they are detrimental.
- b. Relative expression of a gene in a condition is little correlated with its contribution to the fitness, as measured upon deletion
- c. The lack of congruence between expression and fitness cannot be explained by redundancy

Here are some essential points that need to be addressed (the suggested evolution experiment is not mandatory as it can be a long experiment. Yet perhaps after a month of two it would yield results)
Major comments:

1. The lack of correlation between expression level change and essentiality is surprising, but perhaps more expected for lowly expressed genes. While certainly interesting as applied genome-wide, their main finding might be modified, perhaps even reversed if highly expressed genes were considered separately. In fact we do know that very highly expressed genes, such as the lac operon under lactose, are very tightly optimally controlled. I'd suggest thus that the author examine the extent of (dis)agreement between expression fold change and fitness as a function of absolute expression (say in a standard condition). I'd predict that as absolute expression increases the lack of correlation will be replaced by a significant correlation. In simpler terms: redo Fig 3A for highly expressed genes. The amino acid biosynthetic genes might be such examples, if highly expressed in absolute terms.
2. This can certainly wait for a follow up study (i.e. not a requirement for the current paper!): I'd be curious if the authors could use the platform of lab-evolution to evolve the WT strain on a few of the media and see if after several months of evolution the incongruence between expression and essentiality decreases. That would of course require deleting genes again, but perhaps just anecdotal deletions of a sample of genes in each evolved strain (e.g. the extremes in Fig 3A) would suffice. This is related to an issue which can be addressed even now: do we expect conditions to which the organism was exposed in the natural ecology to display lower incongruence compared to completely unfamiliar conditions. I couldn't deduce that from Fig. 1B
3. Related to the above item: What is the growth rate of the wild-type under each condition? Does the extent of incongruence correlate with growth rate? Results could be interesting either way.
4. Still related to the above (and much easier than an evolutionary experiment:) The authors can find a completely non-familiar (to the bugs...) growth medium and see if the incongruence increase even further. If so that would tell us something about the evolutionary/ecological roots of the incongruence between expression and fitness.
5. What is more detrimental to the fitness: not expressing a gene when it's needed, or expressing a

gene when it's not needed. I'd be curious to see if the authors can resolve that (they refer to one of the two cases in "In this case, expressing them when they are not important").

6. Regarding the comment on little anticipatory regulation: this regulatory scheme should only apply between pairs of conditions A and B such that A precedes B in the natural ecology. I'm guessing that for many of the 203 condition pairs examined this is not the case. Hence it's anticipated that anticipatory regulation will not be observed. Actually one of the two exceptions to that - the aerobic to anaerobic switch is very interesting, as they note, especially since (as I just learned) *Shewanella oneidensis* can live both with and without oxygen. In addition, as the authors acknowledged we anticipate anticipation in a very small fraction of the genes, not genome-wide. So it seems that the statement "we found no evidence of anticipatory control" is misleading: although they found some (little?) evidence the current study was not set to find such evidence.

7. One extreme manifestation of the congruence between expression and fitness relates to the absolutely essential genes (which were excluded here for obvious reasons). Yet the authors could still ask about the expression level, or change in expression level, for essential genes in the wild-type. If these genes are highly expressed in the wild type (as was shown in several other species) then the incongruence reported here would be weaker.

Minor comments:

1. The statement "positive fitness indicates that the mutant strain has an advantage and that the gene's activity is detrimental in that condition" could be restricted. There's also the possibility that the gene itself (the DNA segment) is fitness reducing (and the deletion is relieved from this effect), irrespective of the "activity", which is usually ascribed to the gene's product(s).
2. I find it peculiar that there are no detrimental genes on LB, can the authors comment on that?

Reviewer #3 (Remarks to the Author):

In this paper, Price and co-workers explore the correlation between gene expression and its corresponding fitness effect to *S. oneidensis* MR-1 over 15 conditions. They also investigate if their findings can generalize to a dataset of biosynthetic genes in three other bacteria (*E. coli* K-12, *Z. mobilis* ZM-4 and *D. alaskensis* G20). They argue that there is little correlation between how beneficial/detrimental a gene is and its up/down-regulation in different conditions. They also introduce the theory of "indirect control", which states that gene expression responds to environmental signals that are not directly related to the genes' function.

General overview:

I generally agree with many of the statements in this paper, including that the traditional view of gene expression being strongly associated with fitness is not adequate to explain bacterial physiology; that laboratory environments are very different from natural ones; that a lot that is happening in the cell - in gene expression in this case - seems sub-optimal. From an analysis standpoint I feel that they have adequately demonstrated their main argument, that the correlation between gene expression and fitness effect is small at a genome-wide scale. I also think that they addressed this question from multiple angles and they provided a thorough summary and some arguments for the various theories that try to make sense of the heterogeneous data at hand.

General concerns:

1. I have a number of concerns regarding the novelty, validity and usefulness of the conclusions. First, the definition of fitness here (and of course in many other papers), is one-dimensional and restrictive. Is it max/average growth, survival rate, relative abundance (as in this paper)? And in what experimental setting? Even changing the measuring technique (e.g. competition assays vs. microarrays) can have significant impact in measured fitness. How much can your results be generalized if these parameters change? I find statements such the one in page 3 ("We found that 24% of genes are detrimental ... is maladaptive in the laboratory.") to be of little value. These simplifying assumptions and unnatural conditions will of course give "maladaptive", unexplainable results.

2. The fact that there is a mismatch between the fitness effect of a gene and its expression in a laboratory experiment is well-known and documented, in both growing and evolving bacterial populations (the authors reference some of these papers in their introduction). At the same time the

theory of "indirect control" is not really a novel theory as it is similar to the work of U. Alon and others on the lac promoter (Sasson et al., 2012, referenced by the authors). It is not clear how this paper considerably differentiates from previous work and whether any of its results substantially advance the field.

3. The dismissal of the alternative explanations for "suboptimal control" is not well-grounded. For example, in the case of standby control the authors argue that "detrimental" genes should be down-regulated and since they are not, standby control cannot explain "suboptimal control". There are two issues with this argument. First, what if a gene is "detrimental" is solely based on the fitness as defined here (strain abundance), which may not be generally the case if another fitness function is used (or even a different method, such as a competition assay). Second, the "standby control" theory does not argue that the genes with neutral and/or negative fitness contributions should be downregulated - on the contrary these may actually prove to be beneficial in an other environment and exactly because of this uncertainty they can be expressed under these conditions.

Similarly, for anticipatory behavior to be observed, the organism(s) have to be in an environment that has the same correlation-structure as the one that it evolved - which of course is not the case here. A blind search of anticipatory behavior is unlikely to produce results and this is why the authors of both the original (Tagkopoulos et al. 2008) and follow-up (Mitchel et al., 2009) papers targeted natural environments with very clear signal correlations and order. In addition, as the authors also note, the "window of opportunity" for anticipatory behavior to evolve and be advantageous is small, another reason why just looking at a few lab-based non-specific environments will not lead to any results. Furthermore, as the authors point out, weak selection can explain the lack of regulatory control for many proteins, especially when the cost of producing them is small to the cell, or the cost of regulating them high. To this reviewer, these are all theories that complement each other rather than compete.

4. I generally found that there was only a small component of systems biology, as there is a lack of analysis for functional categories and correlations/results within these categories. Also epistatic interactions were not mentioned, except in the case of paralogs, which is an obvious omission. The paper will benefit from a more detailed supplementary information section.

Summary:

While the paper present a solid analysis of the gene expression-fitness relationship, I don't feel that it presents at this point a significant advancement for publication in MSB.

Specific comments:

- Page 4: Define what "strongly-detrimental" genes are before you use it (page 4), the figure implies that these are the genes whose deletion confer a 0.75 fitness increase or higher, relative to the WT.

- Page 4: What is the distribution of experiments in the 38 groups? Is this distribution closer to uniform (i.e. are the bins equally populated)? Otherwise bias can be introduced. Also what is the "typical group of experiments" and how can it have only 1.7% detrimental when 24% of all genes are significantly detrimental?

- Page 5: "This shows that the detrimental activity of most of these genes cannot be explained by optimal standby control: under this model, if genes are expressed because they might be needed after a change in conditions, then they should still be downregulated (Supplementary Figure 1)."

I don't see how you can reject this hypothesis/model based on this result. There might be low down-regulation of detrimental (in that given laboratory environment) genes, which may be beneficial once the environment changes.

- Page 7: The hypothesis that the first and last steps are regulated while the middle genes are constitutively expressed in low-cost pathways is not well argued ("One possibility... constitutively"). Why should this happen, when all genes can be constitutively expressed, thus minimizing fluctuations at negligible cost?

-Page 8: How do the data support that constitutive/growth-regulated genes have a "high cost of

expression" and that this is "not consistent with the cost-of-regulation theory". This is counterintuitive, especially in the case of constitutive genes that the authors found to be less expressed than other genes.

-Page 9: "So this pattern has evolved independently many times" is a rather strong statement from just observing that genes are in different operons.

-Page 11: The first paragraph doesn't convey the message efficiently. How did they come with 4% per generation? what do they mean by "waste of cellular resources" and why doesn't this all-encompassing term explain the growth cost?

-Page 11: "Thus, post-transcriptional regulation cannot explain why much of transcriptional regulation appears to be suboptimal."
However mRNA-Protein correlation can be small (0.3-0.7) and the authors don't discuss the lack of delay when the proteins are present to respond to an imminent stress.

-Page 11: Shutting down genes that don't confer any fitness advantage is not "sub-optimal".

Language:

- The manuscript had almost no typos, but the writing style can be improved considerably.
- As a general comment, the language has to be more formal: "about zero", "mutant strain is sick" (page 4), "genes that are sick only ..." (page 5) are some examples.
- Statements usually lack rigor: "A FEW genes are... better than MOST other strains." (page 4), "in most of the other..."(page

6). Details about statistical tests are lacking in some cases (e.g. "just 302 such cases by chance", page 4) while in other cases the definitions were not clear (e.g. what is a "normal" z-score? are there abnormal ones?)

- Typos: "COG function[al] category", page 4; "and which reaches", suppl. fig.1;

Figures:

1. How many genes are "strongly-detrimental" ? Why the threshold 0.75 was picked here, while in the main text a +/- 0.4 threshold has been selected? From the plot it seems that only 6 are in this category, which lowers the statistical significance of the result considerably. Why are error bars/variance not shown ?

3C. Not clear what the plot shows. Is this the fraction of gene pairs that have a certain "correlation of expression" value and higher/lower as the text describes (page 6)? Then how are the lines not monotonic?

1st Revision - authors' response

05 March 2013

Reviewer #3 raised questions about our method for measuring fitness. We feel that our fitness measurements are robust. In a previous paper, we grew mutant strains individually and showed that their maximum growth rates were strongly correlated with their fitness values from the pooled competition assay (Figure 1F of Deutschbauer et al. 2011). This finding is mentioned and referenced in the growth-phase section of the Results. In the current study, we verified the internal consistency of the fitness values by measuring the fitness of many of our mutants twice, in two different pools. The two fitness values tend to be highly correlated ($r = 0.92$ for the typical experiment). This is described in the Materials and methods. Finally, most of our analyses focus on genes that affect fitness by 4% percent per generation or more. These strong phenotypes are unlikely to be affected by technical issues.

Reviewers #2 and #3 had questions about whether our conditions were unfamiliar to *Shewanella oneidensis* MR-1. As explained in the Discussion, we agree with the reviewers that the artificial nature of the conditions we tested is a limitation of our study, but measuring fitness and gene expression under ecologically realistic conditions would be technically quite challenging. We tested

a wide range of artificial conditions, some of which are more ecologically plausible than others. Conditions that seem likely to be ecologically relevant include aerobic growth on LB or gelatin as a source of peptides; aerobic growth on NAG, lactate, or fatty acids as carbon sources; anaerobic respiration of fumarate with lactate or NAG; or acid stress at pH 6. Two of the stress conditions seem less likely to be ecologically relevant -- 70 micromolar copper(II) is far above the normal concentration in freshwater lakes or lake sediments; and nalidixic acid is an artificial compound and DNA damage stresses in the environment might be quite different. We are not sure about pyruvate and acetate (which MR-1 grows relatively slowly on) or inosine (as MR-1 does not utilize the base portion). We are not sure about the (aerobic) nitrite stress condition -- *S. oneidensis* MR-1 can reduce nitrate to nitrite and then to ammonia (Cruz-Gracia et al., J. Bacteriology 189:656) which should lead it to experience nitrite stress, but it might not experience it under aerobic conditions. Finally, we are not sure about CAS amino acids -- MR-1 grows rapidly on this substrate but this might reflect its adaptation to consuming peptides.

Reviewer #1 had a related question about whether "conditions to which the organism was exposed in the natural ecology to display lower incongruence compared to completely unfamiliar conditions." As shown in Figure 2C, we summarized whether genes that strongly affect fitness are upregulated in a condition by looking at the average change or the D statistic. If we classify those 14 comparisons into those that are more ecologically relevant or not, we do not see any significant difference in either the average change or the D statistic (both $P > 0.25$, t tests). This might imply that all of the conditions are too artificial or that our classification is incorrect. Because this discussion of which conditions are more or less ecologically realistic is highly speculative, it is not included in the revision.

Reviewer #1 and reviewer #3 were concerned that we did not have the right conditions to find anticipatory control and so that we cannot conclude that it does not exist. We agree with the reviewers that we might not have tested the right pairs of conditions, so in the Introduction, we changed "no evidence of anticipatory control" to "little evidence of anticipatory control" (p. 3, penultimate paragraph). However, we did test all pairs of 15 diverse conditions. If anticipatory control were a major reason for why so much of gene expression appears to be non-adaptive, then we think we should have seen more evidence of it.

DETAILED RESPONSES TO REVIEWER #1

"The article would benefit from being more explicit regarding the accuracy of the expression measurements." In the Materials and methods, we report the extent to which genes in the same operon had similar log-levels of expression in our data sets. In the revised Materials and methods, we also report the operon consistency of the log₂ ratios, which ranged from 0.80-0.90 (p. 15, penultimate paragraph). As genes in the same operon will sometimes show different expression responses because of internal promoters (or perhaps because of differing dynamics of RNA degradation), we feel that this demonstrates the high quality of our expression data. The adaptive regulation of most biosynthetic genes in *S. oneidensis* MR-1 (except from nucleotide synthesis genes) also serves as a positive control that our expression measurements are reliable.

To tie the reliability of the expression data more directly to our results, the revision describes three tests of the consistency, within operons, of the instances of apparently non-adaptive regulation. First, Figure 2B compares expression of differentially-fit genes on acetate and lactate and shows that some of them change expression in the "wrong" direction. Among these genes that are in operons, the expression data for other genes confirms the sign of the expression change for 12/18 cases. In the remaining six cases, the expression change of both genes is small (both $|\log_2 \text{ratio}| < 0.5$), which is still consistent with the idea that the operon is not adaptively regulated. See p. 6, second paragraph. Second, the revision reports similar results for validating the genes that are important for fitness on copper chloride but not on lactate, but are down-regulated on copper chloride (p. 6, second paragraph). Third, we reported that many genes have a large change in expression (2-fold up or down) despite not being important for fitness in either condition. We used operon consistency to validate these cases as well -- 80% of the time, the other gene in the operon changes in the same direction and with an absolute log₂ ratio of 0.5. By chance, this would occur only 21% of the time. See p. 6, third paragraph.

Reviewer 1 might also have been concerned about the compendium of 329 gene expression experiments that we used for *S. oneidensis* MR-1. These were conducted with several different platforms, and based on operon correlations, they are of varying quality ($r = 0.22-0.91$, with a median of 0.73). But the sheer number of them should make up for any problems. Figure 3C includes both a positive and a negative control to show that measuring coexpression with this compendium is reliable.

"The authors do not really discuss in depth the rather scandalous question prompted by their own results: if genes are typically not upregulated when they are needed and not downregulated when their expression is detrimental, how come there is so much elaborate expression regulation in bacteria?!" Just because regulation is elaborate need not imply that it arose by selection (Lynch 2007). However, we suspect that the mismatch between laboratory conditions and life in the wild is the more important issue.

Reviewer 1 suggested that "gene balance" might be an important factor in the evolution of bacterial gene regulation and cited a recent review by Birchler & Veitia (2012). This is plausible, but most of the research discussed by Birchler & Veitia concerns multi-cellular organisms, and there is little evidence that gene balance is important for fitness in bacteria. In yeast, phenotypes from underexpressing genes ("haploinsufficiency") seem to be due to a lack of gene product rather than due to disruption of complexes (Deutschbauer et al., *Genetics* 2005). Conversely, the overexpression of genes in yeast is usually deleterious because of regulatory effects rather than because of disrupting the function of a complex (Sopko et al., *Molecular Cell*, 2006). Finally, in bacteria, complexes are often encoded in operons (T. Dandekar et al., *Trends in Biochemical Sciences* 1998), which ensures that expression of the complex will be balanced whether or not the genes are regulated adaptively.

"The article suffers to some extent from quasi-teleological language..." We have made the evolutionary issues more explicit in the Introduction ("these theories try to explain why bacteria with apparently non-adaptive regulation have not been out-competed by other bacteria"; p. 2, third paragraph). We have also changed the language in a few other places. For example, in the discussion in the Results of transcriptional regulation of amino acid and nucleotide utilization in various organisms, we explicitly raise the question of whether transcriptional regulation would "be selected for" instead of whether it should evolve (p. 11, second paragraph). In the discussion of HGT of transcription factors, we replaced "it is difficult" with "selected against" (p. 13, penultimate paragraph).

DETAILED RESPONSES TO REVIEWER #2

Reviewer 2 suggested that highly-expressed genes would show more of a correlation between relative expression and relative fitness. However, in our data for *S. oneidensis* MR-1, high expression does not seem to be a strong indicator of whether the regulation of a gene will appear to be adaptive in the laboratory. Genes that are more highly expressed do tend to show more of an expression-fitness correlation, but the correlation is weak (Spearman rank correlation -0.11 , $P < 1e-9$). Let us consider only genes that have a phenotype ($|\text{fitness}| > 0.75$ in at least one of our matched conditions), and let us define "well-expressed" as expressed 2-fold above the median gene (using the median expression across our 15 matched conditions). As motility genes tend to be well-expressed but do not show an expression-fitness correlation, let us also exclude genes that affect motility (i.e., have motility "fitness" under -0.4). This leaves 76 well-expressed genes that do not affect motility and that have phenotypes in our matched data. Of these 76 genes, 35 are biosynthetic genes that are important for fitness in minimal media; the median expression-fitness correlation of these biosynthetic genes is -0.49 . For the remaining well-expressed genes, the median expression-fitness correlation is just -0.08 , which is significantly weaker than for the well-expressed biosynthetic genes ($P < 0.001$, Wilcoxon rank sum test) and is about the same as for the less-expressed genes that have phenotypes (median -0.07 ; $P > 0.5$, Wilcoxon test). We have added this material to the Results (p. 9, third paragraph).

Reviewer 2 wondered if evolution in the laboratory for a few hundred generations would reduce the incongruence between expression and importance for fitness. It is our impression that for highly expressed genes that are important for fitness, this will happen (e.g., Dekel & Alon 2005). Similarly, we expect that genes that are strongly detrimental to fitness will be disabled or repressed. But for

most proteins that are unnecessary for fitness, we don't expect selection to be strong enough to downregulate them within a few hundred generations.

Could we "find a completely non-familiar (to the bugs...) growth medium and see if the incongruence increase even further?" As discussed above, some of our conditions seem (to us) to be more artificial than others, but it isn't clear that the "more ecological" ones show any more congruence between expression and fitness than the others.

"What is more detrimental to the fitness: not expressing a gene when it's needed, or expressing a gene when it's not needed?" On theoretical grounds, we strongly expect that the former is usually more detrimental. Indeed, in our data, genes are important for fitness much more often than they are detrimental to fitness. For example, in our 15 matched fitness experiments, we had 5,034 cases with $Z < -2.5$ and 1,172 cases with $Z > 2.5$. Based on the reviewer's question, this is now mentioned in the revised Results (p. 4, third paragraph, "For comparison...").

"on little anticipatory regulation: this regulatory scheme should only apply between pairs of conditions A and B such that A precedes B in the natural ecology. I'm guessing that for many of the 203 condition pairs examined this is not the case. Hence it's anticipated that anticipatory regulation will not be observed." We're not sure what pairs of conditions would go together in the environment. When we began this study, we thought we might figure that out from this kind of analysis. However, we tested a variety of conditions and we allowed for all possible pairs between them, and we found little evidence of anticipatory control.

"One extreme manifestation of the congruence between expression and fitness relates to the absolutely essential genes (which were excluded here for obvious reasons)..." Although we did not include essential genes in our study, a reasonable hypothesis is that it would be adaptive for essential genes to be expressed more highly at higher growth rates. Indeed, in *E. coli*, the expression of ribosomal proteins and some other genes that are essential for growth are regulated by growth rate. Alternatively, we imagine that some essential genes might have more of a maintenance function rather than being required for growth per se, they would optimally be expressed constitutively. Also, although the reviewer implied that essential genes should be highly expressed, about 16% of essential genes in MR-1 (based on the list of Deutschbauer et al. 2011) are expressed more weakly than the typical gene; these genes might be constitutively expressed under the cost-of-regulation theory. (Essential genes that seem to be needed at low levels include genes for vitamin synthesis or for DNA replication.) Of the 399 essential genes that we have expression data for, 173 were growth-regulated and 70 were constitutive (together accounting for 61% of the essentials). Another 22 have predicted regulation in RegPrecise, which might indicate a benefit for homeostatic control or might suggest that the requirement for some of these genes is varying even though they are essential for growth in rich media.

"The statement 'positive fitness indicates that the mutant strain has an advantage and that the gene's activity is detrimental in that condition' could be restricted. There's also the possibility that the gene itself (the DNA segment) is fitness reducing (and the deletion is relieved from this effect), irrespective of the 'activity', which is usually ascribed to the gene's product(s)." Our mutants are transposon insertions, so any deleterious activity of the DNA or RNA might still occur. Also, we have more than one insertion within many of our genes, and the data for different insertions in the same gene tends to be very consistent ($r = 0.87$ to 0.96 in our 15 *S. oneidensis* MR-1 fitness experiments). This suggests that the protein is the effective agent. Also, in previous work we complemented 10 of these mutants, including 7 insertions within hypothetical proteins (Deutschbauer et al. 2011). We also note that there was some functional coherence to the detrimental genes. Overall, we believe that the vast majority of our phenotypes are due to expression of the gene itself or, in some cases, due to the polar effects on the expression of downstream genes. Polar effects are not a key issue here as it does not matter which of the genes in an operon are suboptimally controlled. This issue is now addressed in the revised Materials and methods (p. 15, sixth paragraph).

"I find it peculiar that there are no detrimental genes on LB, can the authors comment on that?" Although there are no strongly-detrimental genes in LB, there are 31 detrimental genes on LB (with $Z > 2.5$) and these have some biological consistency: 7 are flagellar genes and 6 others also affect

motility. We are not sure why there are no strongly-detrimental genes on LB. Growth on LB might already be about as fast as MR-1 is physiologically capable of (and no single-gene change might be enough to alter this much). Or, rich conditions might buffer detrimental activity -- i.e, spinning the flagellum and wasting ATP might not matter as much when energy is abundant.

DETAILED RESPONSES TO REVIEWER #3

"[T]he definition of fitness here (and of course in many other papers), is one-dimensional and restrictive. Is it max/average growth, survival rate, relative abundance (as in this paper)?" The reviewer was also concerned that "changing the measuring technique (e.g. competition assays vs. microarrays) can have significant impact in measured fitness" so that our results might not be generalizable. In this work, we defined fitness as the log₂ change in relative abundance during a competitive assay (i.e, growth within a pool of mutants). So we do not understand why the reviewer draws a contrast between a competition assay and our microarray-based assay. One difference is that we examined pools of many mutants instead of head-to-head competition as is traditional, but we do not see why this is important. In previous work, we validated our fitness assay by measuring the maximum growth rate of 48 transposon mutants of *S. oneidensis* MR-1 in isolation and we showed that growth rate and mutant fitness were strongly correlated (Figure 1F of Deutschbauer et al. 2011). In this work, we ensured that our fitness assay was internally consistent by measuring the fitness of many of the strains twice, as many strains are present in both of our pools of mutants. As reported in the Materials and methods, the per-strain fitness values were highly correlated across the two pools, with a correlation of 0.92 in the typical experiment for *S. oneidensis* MR-1. Finally, most of our analyses focus on genes that have a large effect on fitness (either positive or negative) of at least 4% percent per generation. Fitness effects of that size are unlikely to be affected by these technical issues.

The reviewer found "statements such the one in page 3 ('We found that 24% of genes are detrimental ... is maladaptive in the laboratory.')

to be of little value. These simplifying assumptions and unnatural conditions will of course give 'maladaptive', unexplainable results." We agree with reviewer 3 that our experimental conditions are unnatural, and we discuss how this affects the interpretation our results. However, we do not feel that we are making any simplifying assumptions. Well-shaken growth in an artificial medium is the standard setup of bacterial physiology, and we are probably the first to show that so many bacterial genes are in fact detrimental under such conditions. Furthermore, we showed that genes do not tend to be down-regulated when they are detrimental.

The "mismatch between the fitness effect of a gene and its expression in a laboratory experiment is well-known and documented, in both growing and evolving bacterial populations (the authors reference some of these papers in their introduction)." None of the previous works include genome-wide data on the extent to which this mismatch occurs except for a brief mention in our previous paper (Deutschbauer et al. 2011).

"[The theory of "indirect control" is not really a novel theory as it is similar to the work of U. Alon and others on the lac promoter (Sasson et al., 2012, referenced by the authors)." Our manuscript discusses the similarities and differences between our theory and that of Sasson et al. One key difference is that Sasson et al.'s theory explains minor fluctuations away from sensible control, but does not predict the total absence of adaptive control for most genes (at least in the laboratory) that we have demonstrated. (Indeed, Sasson et al. study the mostly adaptively-regulated lac promoter and variants thereof.) An expanded discussion of the differences between our theory and that of Sasson et al. is included in the revised Discussion (p. 14, third paragraph). Another key difference is that, in our view, the limited number of sensors and regulators is likely to be a major constraint on the evolution of adaptive regulation. This constraint is not mentioned by Sasson et al. and is rarely mentioned in previous works on this topic.

Reviewer 3 had two criticisms of our argument that standby control cannot explain the patterns that we observed. First, "the authors argue that 'detrimental' genes should be down-regulated and since they are not, standby control cannot explain 'suboptimal control.'" There are two issues with this argument. First, what if a gene is 'detrimental' is solely based on the fitness as defined here (strain abundance), which may not be generally the case if another fitness function is used (or even a different method, such as a competition assay)." As discussed above, we do use a competition assay, and we focus on genes that have large effects on fitness that should be insensitive to these technical

issues.

Second, "the 'standby control' theory does not argue that the genes with neutral and/or negative fitness contributions should be downregulated - on the contrary these may actually prove to be beneficial in an other environment and exactly because of this uncertainty they can be expressed under these conditions." The standby control theory explains why they can be expressed at functionally significant levels, but given the cost-benefit tradeoff, the benefit is less, so they should still be downregulated. A more detailed version of this argument is given in Supplementary Figure 1. We also note that we identified many genes that reduce the growth rate by ~4% per generation. The odds of a switch to a different condition in which that gene is beneficial would need to be very high (above 4% per generation) to justify such a high cost of standby control.

Reviewer #3 criticized our analysis of anticipatory control, stating that "looking at a few lab-based non-specific environments will not lead to any results". We did a systematic analysis with 15 diverse conditions, and we considered virtually all pairs of those conditions (>200 pairs tested). When we began this work we had thought that some of them would anticipate each other (e.g. lactate accumulation might indicate competition with fermenting bacteria and hence predict anaerobic growth or a future drop in pH). But we did not find evidence of this.

Reviewer #3 also stated that various other theories (standby control, weak selection, anticipatory control) may be complementary to our model of indirect control, rather than being competing theories. We agree with this view and Figure 5 illustrates that a significant number of genes can be explained by the cost-of-regulation theory. However, it seems unlikely that the other theories are operating on a large enough scale to explain why so few genes are adaptively regulated (at least under laboratory conditions).

"I generally found that there was only a small component of systems biology, as there is a lack of analysis for functional categories and correlations/results within these categories." The manuscript highlights two results for functional categories, (i) biosynthetic genes (except nucleotide synthesis genes) tend to be under adaptive control in *S. oneidensis* MR-1, and (ii) motility genes tend to be detrimental. We also discuss the functional enrichment (or lack thereof) for constitutive and growth-regulated genes.

"[E]pistatic interactions were not mentioned, except in the case of paralogs, which is an obvious omission." We suspect that reviewer 3 is concerned about genetic redundancy (and not alleviating epistasis). We tested paralogs because they are often believed to be redundant. Unfortunately, we do not see a good way to test the issue more broadly. But genetic redundancy cannot explain why so many genes are detrimental to fitness in the laboratory or why they are not downregulated when they are detrimental. This is mentioned briefly in the revised Results section titled "Relative expression is little correlated with fitness" (p. 7, first paragraph).

Also, we do not expect genetic redundancy to affect our analysis. Consider a pair of genes that are genetically redundant -- mutations in either gene has too small an effect on fitness for us to measure, but a double mutation would be very sick. If the pair is redundant in all conditions then single mutants will not have phenotypes and both genes will be excluded from most of our analyses. If the pair is redundant in some conditions but not others, then it would seem adaptive for a gene to be more highly expressed when it is not redundant than when it is. In particular, in a condition where the genes are redundant, the cell could probably save resources by expressing one of them at lower levels, as compared to when their activities are not redundant. So we expect that adaptive control would often lead to an expression-fitness correlation for sometimes-redundant genes.

"The paper will benefit from a more detailed supplementary information section." Unfortunately, reviewer 3 did not give us any suggestions as to what material should be added to the supplementary information. All of our data is available from public databases (Gene Expression Omnibus for microarrays and RNA-Seq; MicrobesOnline for fitness data). Also, we have a web site for the paper to make it easier for other scientists to replicate our analyses (<http://genomics.lbl.gov/supplemental/exprVfitness2012/>).

"Define what 'strongly-detrimental' genes are before you use it (page 4), the figure implies that these are the genes whose deletion confer a 0.75 fitness increase or higher, relative to the WT." That is

correct, and based on the reviewer's question, we have revised the text (p. 4, third paragraph).

"What is the distribution of experiments in the 38 groups? Is this distribution closer to uniform (i.e. are the bins equally populated)? Otherwise bias can be introduced. Also what is the 'typical group of experiments' and how can it have only 1.7% detrimental when 24% of all genes are significantly detrimental?" The size of the groups of experiments is not uniform but we do not see how this introduces bias. We do not make any claims based on the tendency of genes to be detrimental in one group of experiments versus another. The "typical" rate of 1.7% reports the median, across the 38 groups, of the fraction of genes that are detrimental in that group. If there were no overlap across groups of experiments as to which genes were detrimental, then we would have roughly $1.7\% \times 38 = 65\%$ of genes detrimental. Not surprisingly, there is considerable overlap, so "only" 24% of genes were significantly detrimental. As this material is not central to our argument and seems to be confusing, we have removed it from the Results (p. 4, penultimate paragraph).

"This shows that the detrimental activity of most of these genes cannot be explained by optimal standby control: under this model, if genes are expressed because they might be needed after a change in conditions, then they should still be downregulated (Supplementary Figure 1). I don't see how you can reject this hypothesis/model based on this result. There might be low down-regulation of detrimental (in that given laboratory environment) genes, which may be beneficial once the environment changes." We are not sure if we understand "low down-regulation." Reviewer 3 might be arguing that the detrimental genes are being downregulated, but by such a small amount that we cannot measure it. Given that we examined expression measurements for over 1,000 cases in which genes were detrimental, this would be a subtle effect indeed.

"The hypothesis that the first and last steps are regulated while the middle genes are constitutively expressed in low-cost pathways is not well argued... Why should this happen, when all genes can be constitutively expressed, thus minimizing fluctuations at negligible cost?" In the theory of Wesselly 2011, the regulation must control the flux, and any reduction in the amount of protein that is needed is a side benefit. This was poorly worded in our original submission and we have revised the text to better explain this (p. 7, last paragraph).

"How do the data support that constitutive/growth-regulated genes have a 'high cost of expression' and that this is 'not consistent with the cost-of-regulation theory'. This is counterintuitive, especially in the case of constitutive genes that the authors found to be less expressed than other genes." We found that 28% of constitutive genes were measurably detrimental to fitness in some laboratory conditions, which shows that improved regulation could have a significant fitness advantage. This is not consistent with the cost-of-regulation theory. Also, only about half of the constitutive genes are weakly expressed and not detrimental to fitness. So, it seems to us that the cost-of-regulation theory can explain some of the constitutive genes (as shown in Figure 5) but not the others. Our results for growth-regulated genes are even less consistent with the cost-of-regulation theory as they tend to be highly expressed and more of them are sometimes detrimental to fitness. Based on the concerns of reviewer 3, we have edited this text, which we hope makes it clearer (p. 9, first paragraph).

"So this pattern has evolved independently many times' is a rather strong statement from just observing that genes are in different operons." These genes are in 11 different operons, which suggests to us that these 11 different promoters (for genes related to nucleotide synthesis) independently evolved such that they are down-regulated in minimal media. We do not understand the reviewer's concern.

"How did they come with 4% per generation? what do they mean by 'waste of cellular resources' and why doesn't this all-encompassing term explain the growth cost?" A change in \log_2 abundance of +0.4 across seven generations means that the relative abundance increases by a factor of $2^{0.4/7} \approx 1.04$ each generation, or a 4% per generation increase in abundance. This is mentioned briefly in the revised Discussion (p. 12, second paragraph). We have also revised the surrounding text to make it clearer -- the 4% is the growth advantage of a mutant, and the cost of making unneeded protein seems too small to explain it.

This reviewer questioned our claim that "post-transcriptional regulation cannot explain why much of transcriptional regulation appears to be suboptimal", stating that "mRNA-Protein correlation can be

small (0.3-0.7)." The key issue is whether changes in mRNA levels lead to corresponding changes in protein levels. We do not know of any reliable large-scale data for this question for any bacterium. (Some studies have looked at the correlation between the absolute abundance of protein and mRNA in a single condition, which might be what the reviewer is referring to.) Because the vast majority of known gene regulation in bacteria is transcriptional, it seems unlikely that post-transcriptional mechanisms are a big part of the explanation. We also note that translational repression is expected to decrease transcript stability and hence to reduce mRNA levels. In other words, post-transcriptional regulatory mechanisms would often affect mRNA levels and would show up in our data. This is mentioned briefly in the revised Discussion (p. 12, third paragraph).

"[T]he authors don't discuss the lack of delay when the proteins are present to respond to an imminent stress." This is taken into account under the theory of standby control, which we test extensively. Indeed, the key benefit of standby expression is this lack of delay. We hope that this is better explained in the revised Introduction (p. 2, third paragraph).

"Shutting down genes that don't confer any fitness advantage is not 'sub-optimal'." (p. 11). If a gene's expression goes strongly up and down, across conditions in which the gene is not important for fitness, then this suggests that the gene's expression is responding to extraneous factors, as in our model of indirect control. Furthermore, although downregulating the gene in some of the conditions could be adaptive, these genes are also upregulated in some conditions, which appears to be sub-optimal.

"As a general comment, the language has to be more formal: 'about zero', 'mutant strain is sick' (page 4), 'genes that are sick only ...' (page 5) are some examples." "About zero" is correct -- the normalization is designed to give wild-type a fitness of zero, but we cannot exactly reach this goal. The term "sick" used for brevity and is introduced at the beginning of the Results section so we think this should be clear. We have changed some other phrases that might be difficult to understand, e.g. "one wonders why" to "it is not clear why" (p. 4, third paragraph) or "We also wondered whether" to "To test whether" (p. 5, second paragraph).

"Statements usually lack rigorousness: 'A FEW genes are... better than MOST other strains.' (page 4), 'in most of the other...' (page 6)." In the first case, the quantitation is given in the referred-to figure. In the second case, we revised "in most of..." to give the full range of values (p. 6, third paragraph, "For all of our comparisons..."). We have also added the exact numbers in other locations in the text. For example, we now report the number of protein-coding genes absent from our data set that are essential, short, or repetitive (p. 4, second paragraph). We now report the fraction of time that the Z score is significant if $|\text{fitness}|$ is above 0.75 (p. 18, fourth paragraph).

"Details about statistical tests are lacking in some cases (e.g. 'just 302 such cases by chance', page 4) while in other cases the definitions were not clear (e.g. what is a 'normal' z-score? are there abnormal ones?)." In the standard normal distribution, values of 2.5 or above occur 0.62% of the time. We examined 15 conditions and 3,247 genes, and $15 \times 3247 \times 0.0062 \approx 302$, so we expect 302 such cases by chance. In the revised Results (p. 4, third paragraph), we briefly explain the origin of the Z scores (which are described in more detail in the Materials and methods) and this computation.

Figure 1 -- "How many genes are 'strongly-detrimental' ? Why the threshold 0.75 was picked here, while in the main text a +/- 0.4 threshold has been selected? From the plot it seems that only 6 are in this category, which lowers the statistical significance of the result considerably." Fitness above 0.75 is the threshold for "strongly detrimental genes" (highlighted in the original Figure 1A), while $Z > 2.5$ is the threshold for "potentially-significant detrimental activity" (summarized in Figure 1B). To try to reduce confusion, the revised Figure 1A color-codes the significantly detrimental or beneficial genes ($|Z| > 2.5$) and has a vertical line to emphasize the strongly-detrimental genes with fitness values above 0.75. Also, Figure 1A and 1B are based on fitness values from a single experiment. When we average multiple experiments, we use a different statistical test and we use a lower threshold for fitness, namely fitness above 0.4.

Figure 1 -- "Why are error bars/variance not shown?" -- We have added confidence intervals to Figure 1B.

Figure 3C -- "Not clear what the plot shows. Is this the fraction of gene pairs that have a certain

'correlation of expression' value and higher/lower as the text describes (page 6)?" Then how are the lines not monotonic?" This figure shows distributions, not cumulative proportions. We hope that the revised caption for Figure 3C is clearer.